# WNT11-FZD7-DAAM1 signalling supports tumour initiating abilities and melanoma amoeboid invasion

Irene Rodriguez-Hernandez [1,2], Oscar Maiques[1,2], Leonie Kohlhammer[1,2], Gaia Cantelli[2,10], Anna Perdrix-Rosell[1,2,3], Joanne Monger[1], Bruce Fanshawe[2,4], Victoria L. Bridgeman[3], Sophia N. Karagiannis [5], Rosa M. Penin[6], Joaquim Marcolval[7], Rosa M. Marti [8], Xavier Matias-Guiu [9], Gilbert O. Fruhwirth [4], Jose L. Orgaz[1,2,11], Ilaria Malanchi [3] & Victoria Sanz-Moreno [1,2✉]

Melanoma is a highly aggressive tumour that can metastasize very early in disease progression. Notably, melanoma can disseminate using amoeboid invasive strategies. We show here that high Myosin II activity, high levels of ki-67 and high tumour-initiating abilities are characteristic of invasive amoeboid melanoma cells. Mechanistically, we find that WNT11-FZD7-DAAM1 activates Rho-ROCK1/2-Myosin II and plays a crucial role in regulating tumour-initiating potential, local invasion and distant metastasis formation. Importantly, amoeboid melanoma cells express both proliferative and invasive gene signatures. As such, invasive fronts of human and mouse melanomas are enriched in amoeboid cells that are also ki-67 positive. This pattern is further enhanced in metastatic lesions. We propose eradication of amoeboid melanoma cells after surgical removal as a therapeutic strategy.

[1] Barts Cancer Institute, Queen Mary University of London, John Vane Science Building, Charterhouse Square, London EC1M 6BQ, UK. [2] Randall Division of Cell and Molecular Biophysics, New Hunt's House, Guy's Campus, King's College London, London SE1 1UL, UK. [3] Tumour Host Interaction Laboratory, The Francis Crick Institute, 1 Midland Rd, London NW1 1AT, UK. [4] Department of Imaging Chemistry and Biology, School of Biomedical Engineering and Imaging Sciences, Kings' College London, London SE1 7EH, UK. [5] St John's Institute of Dermatology, School of Basic & Medical Biosciences, King's College London and NIHR Biomedical Research Centre at Guy's and St Thomas' Hospitals and King's College London, London SE1 9RT, UK. [6] Department of Pathology, Hospital Universitari de Bellvitge, IDIBELL, l'Hospitalet de Llobregat, 08908 Barcelona, Spain. [7] Department of Dermatology, Hospital Universitari de Bellvitge, IDIBELL, l'Hospitalet de Llobregat, 08908 Barcelona, Spain. [8] Department of Dermatology, Hospital Universitari Arnau de Vilanova, University of Lleida, IRB Lleidal, CIBERONC, 25198 Lleida, Spain. [9] Department of Pathology and Molecular Genetics, Hospital Universitari Arnau de Vilanova, University of Lleida, IRB Lleida, CIBERONC, 25198 Lleida, Spain. [10]Present address: Department of Medicine, Division of Hematologic Malignancies and Cellular Therapy, Duke University, Durham, NC, USA. [11]Present address: Instituto de Investigaciones Biomedicas 'Alberto Sols', CSIC-UAM, 28029 Madrid, Spain. ✉email: v.sanz-moreno@qmul.ac.uk

Malignant melanoma is one of the most aggressive cancers responsible for more than 80% of skin cancer-related deaths. Melanoma is a non-epithelial tumour originating in melanocytes, the pigment-producing cells derived from the neural crest (NC). NC is a highly migratory and multipotent cell population that undergoes epithelial to mesenchymal transition (EMT) during embryogenesis[1]. The highly metastatic and therapy-resistant features of melanoma have been linked to its NC-like phenotype with stemness properties[1–4]. Melanomas are very heterogeneous and comprise different subpopulations of cells with stem cell-like properties[5,6]. These cells are required to initiate and sustain tumour growth, therapy resistance and subsequent tumour relapse[4,5].

Migratory melanoma cells display high cellular plasticity, shifting between elongated and amoeboid modes of motility to adapt to changing microenvironments[7–10]. Both migratory modes require Myosin II activity to a certain extent, but amoeboid cells are characterized by hyper-activation of Rho-ROCK1/2 signalling[9–12]. ROCK1/2 activation downstream of RhoA/C results in decreased myosin phosphatase activity and increased MLC2 phosphorylation and Myosin II activity. Amoeboid cells migrate faster in vivo and are enriched in the invasive fronts of primary tumours and in secondary lesions[7,10,12–16]. On the other hand, in epithelial tumours, acquisition of a migratory and invasive phenotype via EMT has been linked to the increase in stem cell-like properties[17]. In melanoma, switching from proliferative to invasive states has been associated with melanoma progression and metastasis[18,19]. Although this phenotype switching resembles an EMT-like behaviour, the connection with tumour-initiating abilities in melanoma remains elusive[20,21]. Furthermore, whether ROCK-driven amoeboid invasive behaviour is associated with stem cell-like properties is not clear.

In this study, we show how amoeboid invasive melanoma cells with high Myosin II activity harbour tumour-initiating abilities. We find that ROCK1/2 regulates EMT-related genes that, in return, control amoeboid features. WNT11-FZD7-DAAM1 activates Rho-ROCK1/2-Myosin II to control tumour-initiating potential, local invasion and distant metastasis formation. We show that the invasive fronts of human primary melanomas are enriched in amoeboid invading and proliferating cells expressing non-canonical Wnt and cancer stem cell markers.

## Results

**Amoeboid cells support tumour initiation in melanoma in vitro and in vivo**. To explore if amoeboid melanoma cells express genes related to NC development or stem cell-like features, single-sample gene set enrichment analysis (ssGSEA) was performed in a published transcriptional signature for amoeboid melanoma cells[10]. Specifically, the transcriptomes of amoeboid A375M2 melanoma cells were compared to A375M2 cells treated with ROCK1/2 inhibitors (ROCKi) or blebbistatin, a direct Myosin II inhibitor, or compared to intrinsically less amoeboid and less metastatic A375P melanoma cells with lower Myosin II activity[10,12,15]. ssGSEA analysis revealed that A375M2 cells were enriched in different gene signatures that confer essential attributes of stem cell-like properties, including NC genes, cancer stem cell or embryonic stem cell signatures (Fig. 1a).

Next, we assessed the in vivo tumour-initiating potential of limiting dilutions (500,000, 50,000 and 5,000 cells) of A375M2 and A375P cells subcutaneously injected into immunodeficient NOD/SCID/IL2Rγ−/− (NSG) mice and using extreme limiting dilution analysis (ELDA)[22]. Amoeboid A375M2 cells were more efficient in tumour initiation, with a significant difference in tumour-initiating frequency (TIF) (Fig. 1b), and showed increased tumour growth in all conditions compared to A375P cells (Fig. 1c).

Enrichment in rounded cells (Fig. 1d) and Myosin II activity, as measured by phosphorylated MLC2 (p-MLC2) levels (Fig. 1e), were observed at the invasive front (IF) of A375M2 tumours compared with tumour body (TB), while decreased cell rounding and Myosin II levels were found in A375P tumours (Fig. 1d, e). Importantly, we also observed an increase in amoeboid features in IFs of A375P tumours compared to TBs, although less pronounced (Fig. 1d, e). To further investigate the heterogeneity of Myosin II levels within the tumours, Myosin II activity was scored from 0 (low) to 3 (very high) based on p-MLC2 intensity. A375M2 tumours showed an increase in cells with very high Myosin II in the IF (Supplementary Fig. 1a). High ki-67 levels have been associated with the aggressiveness of cutaneous melanoma[23]. Although no differences were observed in cell numbers in vitro after 7 days in culture (Supplementary Fig. 1b), A375M2 tumours showed a higher proliferation index in vivo, as evidenced by ki-67 staining (Supplementary Fig. 1c). Interestingly, IFs of all tumours were enriched in ki-67 proliferative cells. These data suggest that amoeboid cells with intrinsically high Myosin II activity are also proliferative and promote tumour initiation in vivo.

We next investigated in vitro self-renewal capacity of melanoma cells in low adherent conditions. We introduced another pair of melanoma cell lines, WM983B (metastatic, rounded-amoeboid and high Myosin II) and WM983A (primary tumour, elongated and low Myosin II)[15,24] derived from the same patient. Using these two models, we performed serial sphere passages of elongated melanoma cells with low levels of Myosin II[12,15,24] (A375P and WM983A) (Supplementary Fig. 1d). Serial passaging resulted in cells with increased melanosphere formation abilities over time (Fig. 1f). Although tumour-initiating cells are described to be in a slow proliferative state[25], sub-populations of proliferating stem cells have also been found in some tumours[26]. Immunohistochemical analysis of melanospheres revealed an increase in Myosin II activity (Fig. 1g) and higher percentage of ki-67 positive cells (Fig. 1h) with increasing passage number. Morphology of cells from adherent conditions and of dissociated single cells from serially passaged spheres was also assessed on collagen I matrices to recapitulate the dermal environment[7,8,10,12,14]. Importantly, serial passages resulted in an enrichment of rounded cells (Fig. 1i) with high Myosin II levels (Fig. 1j) and increased blebbing (Fig. 1k). Increased self-renewal ability was, therefore, associated with increased amoeboid features. Although the enrichments were less pronounced, similar results were obtained when serial sphere passages were performed in cells in an already amoeboid phenotype (A375M2 and WM983B) (Supplementary Fig. 1e–j). Moreover, MLC2-GFP was transduced into WM983A cells (Supplementary Fig. 1k, l) and this induced increased melanosphere formation, increased cell rounding and increased Myosin II activity (Supplementary Fig. 1m–o).

Overall, these data show that amoeboid cells are more tumourigenic and sustain self-renewal and tumour initiation in melanoma.

**EMT genes regulated by ROCK1/2 control amoeboid invasive features**. Melanoma is a non-epithelial tumour, hence acquisition of invasive features is not considered a canonical EMT[27]. Nevertheless, EMT gene expression has been associated with the acquisition of stem cell-like properties. ssGSEA analysis in our signature for amoeboid melanoma cells[10] revealed that amoeboid A375M2 cells were enriched in both EMT and metastasis-related gene signatures (Fig. 2a). Of note, we also found that amoeboid cells express genes from NC and invasive signatures associated with drug resistance[4,24] (Figs. 1a, 2a). The generated amoeboid transcriptomes were obtained after 24 h of ROCK1/2-Myosin II

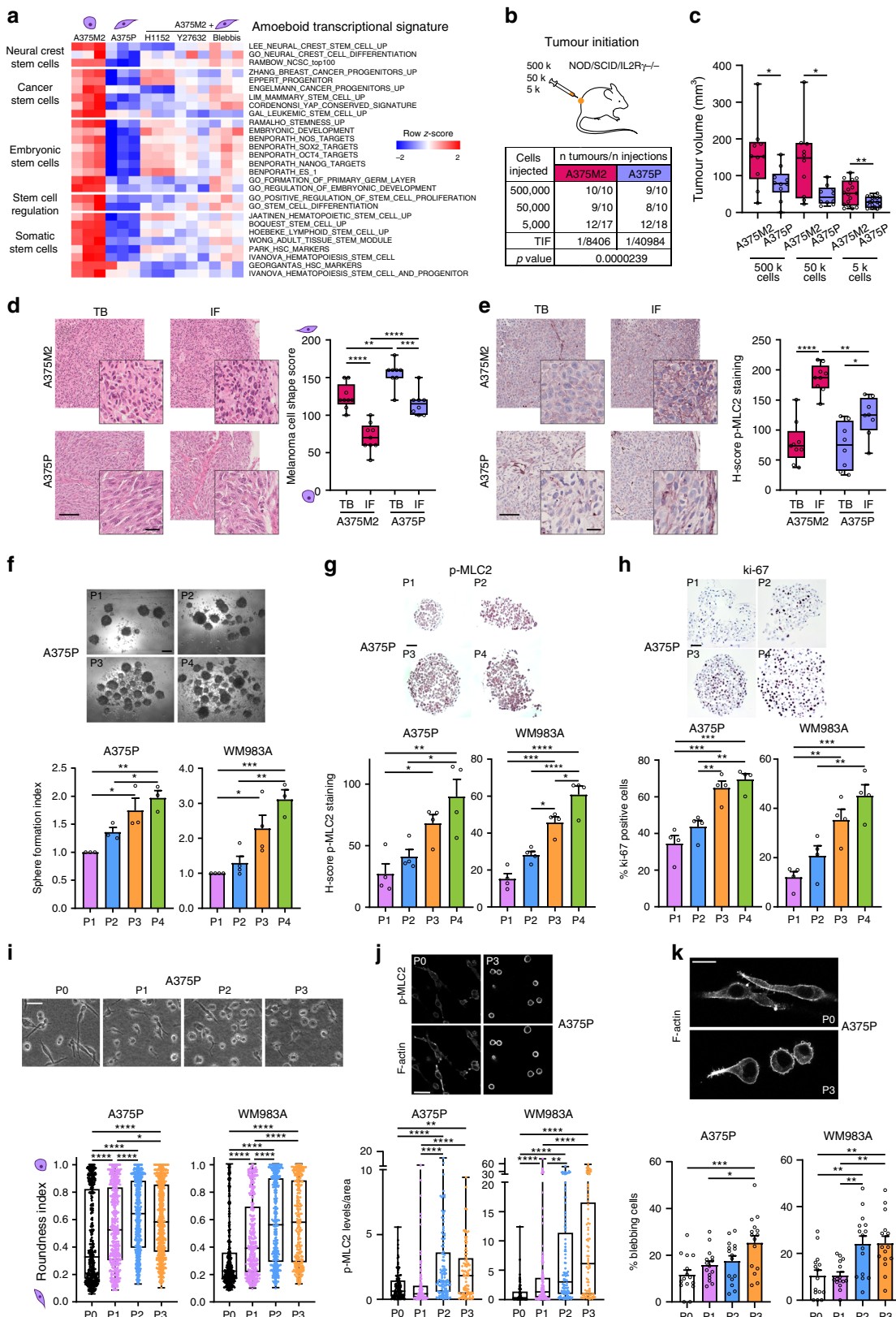

inhibition[10]. To assess more direct effects of ROCK1/2 inhibition, we performed an EMT-directed qPCR array in A375M2 and WM1361 melanoma cells (BRAF$^{V600E}$ or NRAS$^{Q61L}$-driven melanoma, respectively, to account for the main melanoma oncogenic drivers) treated with ROCKi for only 4 h. At this time point, loss of cell rounding and decreased Myosin II activity were already observed (Fig. 2b). Interestingly, ROCKi decreased expression of 72% and 59% of EMT-related genes in A375M2 and WM1361 cells, respectively (Fig. 2c and Supplementary Fig. 2a). ROCKi treatment resulted in significantly reduced expression of key EMT regulators, such as SNAI family members and the mesenchymal markers N-cadherin, Vimentin and Fibronectin

**Fig. 1 Amoeboid cells support tumour initiation in melanoma in vitro and in vivo. a** Heatmap displaying enrichment scores for differentially expressed stem cell-related signatures in amoeboid A375M2 cells compared to A375P cells or to A375M2 cells treated with ROCK1/2 inhibitors (ROCKi) (H1152 and Y27632) or blebbistatin using single-sample Gene Set Enrichment Analysis (ssGSEA). **b, c** Limiting dilution assay estimating **b** tumour-initiating frequency (TIF) and **c** tumour volume of A375M2 and A375P cells when injected at different dilutions (500,000, 50,000 and 5,000 cells) into NOD/SCID/ IL2Rγ−/− (NSG) mice (Number of tumours per condition indicated in table). TIF was determined using ELDA. **d, e** Representative images (left) and quantification (right) of **d** melanoma cell shape score and **e** H-score of p-MLC2 staining in tumour body (TB) and invasive front (IF) of A375M2 ($n = 9$) and A375P ($n = 8$) tumours from 50,000 cells' condition from (**b**). Scale bar, 100 μm; inset, 25 μm. **f** Representative phase-contrast images (top) and quantification of sphere formation index (bottom) of A375P ($n = 3$) and WM983A cells ($n = 4$) serially passaged. Scale bar, 250 μm. **g, h** Representative images (top) and quantification (bottom) of **g** H-score of p-MLC2 staining and **h** ki-67 positive cells in A375P and WM983A spheres serially passaged ($n = 4$). Scale bar, 50 μm. **i–k** Representative **i** phase-contrast and **j, k** confocal images (top) and quantification (bottom) of **i** cell morphology (>250 cells pooled from $n = 3$), **j** p-MLC2 immunofluorescence signal normalized by cell area (>80 cells pooled from $n = 4$) and **k** percentage of blebbing cells (5 fields of view per experiment, >75 cells per experiment, $n = 3$) of individual A375P and WM983A cells from adherent conditions (P0) and from dissociated cells from spheres serially passaged (P1–P3) on collagen I matrix. Scale bar, **i, j** 50 μm and **k** 20 μm. **c, i, j** Box limits show 25th and 75th percentiles, the horizontal line shows the median, and whiskers show the minimum and maximum range of values. **f–h, k** Graphs show mean ± s.e.m. **f–k** $n$ means number of independent biological experiments. **c** two-tailed $t$-test. **d–h, k** One-way ANOVA with Tukey post-hoc test. **i, j** Kruskal–Wallis with Dunn's multiple comparison test. For all graphs, $*p < 0.05$, $**p < 0.01$, $***p < 0.001$, $****p < 0.0001$. The exact significant $p$ values for $*p$, $**p$ and $***p$ are provided in Supplementary Table 1. Mouse schematic in this figure was created using Servier Medical Art templates licensed under a Creative Commons Attribution 3.0 Unported License (https://smart.servier.com).

(Fig. 2c). These data show that ROCK1/2 are positive regulators of EMT-related gene expression, independently of melanoma oncogenic background.

To identify key signalling pathways associated with this gene signature, network enrichment analysis was performed. The most significantly enriched network was centred on Wnt and TGFβ signalling, while the second one encompassed NF-κB and STAT3 signalling (Supplementary Fig. 2b, c). We selected genes from these top networks with no prior link to amoeboid migration (*WNT11*, *WNT5B*, *AHNAK*, *TCF4* and *CAV2*) and investigated if they could regulate amoeboid features. As positive control, *SERPINE1* was selected as a gene from our top network previously described to support amoeboid behaviour[28]. Amoeboid cells are characterised by rounded morphology and high Myosin II activity, so we first measured these two features on WM1361 cells grown on collagen I matrices. Reducing expression of *SERPINE1*, *WNT11*, *WNT5B*, *AHNAK* and *TCF4* led to both decreased cell rounding and decreased Myosin II activity (Fig. 2d–f). During melanoma progression, melanoma cells detach from the surrounding keratinocytes in order to invade the dermis. Amoeboid melanoma cells adhere less to keratinocytes and migrate more efficiently through collagen I compared with elongated cells with lower Myosin II[7]. We found that *SERPINE1*, *WNT11*, *WNT5B* and *CAV2* depletion increased adhesion of WM1361 cells to keratinocytes (Fig. 2g), while depletion of *SERPINE1*, *WNT11*, *WNT5B* and *AHNAK* reduced 3D invasion through collagen I matrices (Fig. 2h). In summary, *WNT11*, *WNT5B* and *SERPINE1* (our positive control) had an impact in all assays (Fig. 2i). We next validated the role of non-canonical Wnt ligands in regulating amoeboid features found in our screens. Using three different RNAi to reduce *WNT11* expression, we measured decreased cell rounding, Myosin II activity and reduced invasion (Supplementary Fig. 2d–g) in both A375M2 and WM1361 cells. Similar results were obtained after *WNT5B* knockdown (Supplementary Fig. 2h–j).

These results show that non-canonical Wnt ligands support melanoma amoeboid invasion independently of oncogenic background.

**Non-canonical Wnt ligands support melanosphere formation and amoeboid behaviour**. Non-canonical Wnt ligands are regulated by ROCK1/2 and in return these ligands control amoeboid features, generating a positive feedback loop (Fig. 2c–i). ROCK1/2-Myosin II has multiple cellular functions including cell proliferation, migration or invasion[7,9–11,29–31], while EMT has

been associated with increased stem cell-like properties[17]. We first explored whether ROCK1/2-Myosin II sustains in vitro tumour initiation by blocking the entire transcriptional programme using different ROCKi (H1152 and GSK269962A). Expression of stem cell-related genes (Fig. 3a and Supplementary Fig. 3a), cell rounding and p-MLC2 levels (Supplementary Fig. 3b, c) were decreased after treatment. Interestingly, using several amoeboid cell lines, one single treatment with ROCKi impaired melanosphere formation (Fig. 3b and Supplementary Fig. 3d). Of note, in vitro 2D cell viability was reduced after 7 days (Fig. 3c and Supplementary Fig. 3e). Similar results were obtained blocking Myosin II directly with blebbistatin (Fig. 3b, c). These effects were further confirmed to be ROCK1/2-Myosin II dependent, since specific silencing of both ROCK genes (*ROCK1/ 2*) or both MLC2 genes (*MYL9* and *MYL12B*) reduced melanosphere formation in A375M2 and WM1361 cells (Fig. 3d and Supplementary Fig. 3f, g). 2D cell viability was decreased after 7 days (Fig. 3e), while no effects were measured after 3 days (Fig. 3f). These data show that we can transiently reduce Myosin II levels without affecting melanoma cell viability but Myosin II long-term depletion compromises cell survival.

We, therefore, next explored whether ROCK1/2-Myosin II sustains tumour initiation via regulation of Wnt ligands and independently of other functions. Interestingly, reducing levels of *WNT11* and *WNT5B* resulted in reduced melanosphere formation (Fig. 3g, h), while in vitro 2D cell viability was unaffected after 7 days (Fig. 3i, j). These data suggest that Wnt ligands support both amoeboid features and specifically in vitro melanosphere formation. On the other hand, Wnt signals can be produced by tumour cells as well as by stromal cells in the tumour microenvironment[32]. Elongated A375P cells treated with either WNT11 or WNT5B ligands increased cell rounding (Fig. 3k). These results indicate that melanoma cells are capable of regulating amoeboid features in response to non-canonical Wnt ligands in an autocrine and paracrine manner.

**FZD7 downstream of WNT11 supports melanosphere formation and amoeboid invasion via DAAM1**. Non-canonical Wnt ligands can signal through different Fzd receptors and co-receptors. Using our microarray data for amoeboid melanoma cells[10], we found that most of the receptors for WNT11 and WNT5B were overexpressed in amoeboid A375M2 cells (Fig. 4a). Since amoeboid cell content is enriched in advanced stages of melanoma progression[7,8,10,12], we used publicly available data[33–35] to investigate receptor expression for non-canonical

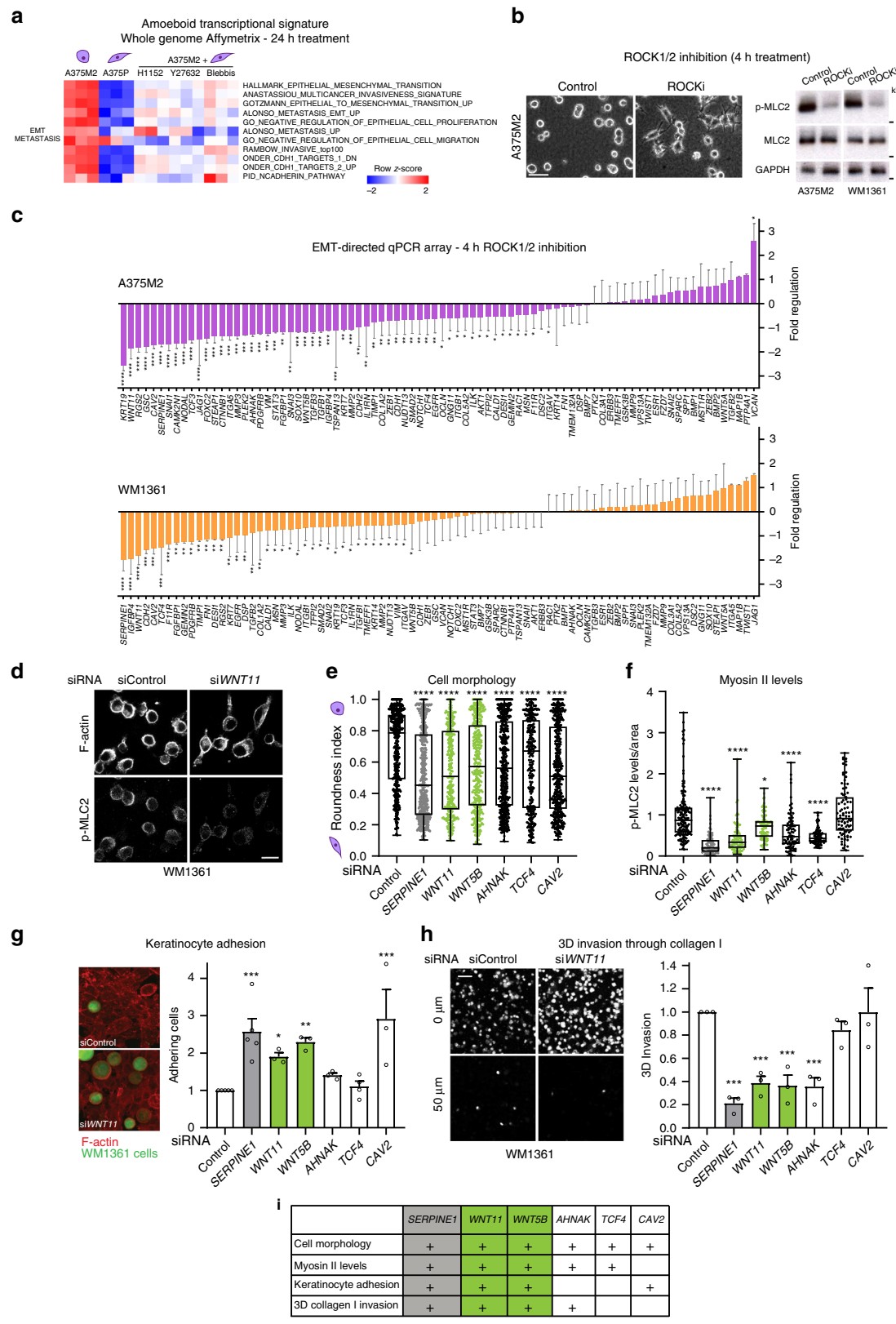

Wnt ligands in primary and metastatic human melanomas (Fig. 4b). In particular, *WNT11*-cognate receptor *FZD7* and its co-receptor *RYK* were both consistently upregulated in metastatic melanomas in most of the studies. Importantly, *FZD7* silencing resulted in the strongest reduction of melanosphere formation (Fig. 4c), while depletion of either *FZD7* or *RYK* did

not affect in vitro 2D cell viability (Fig. 4d). *FZD7* silencing also resulted in loss of cell rounding (Fig. 4e) and decreased Myosin II activity (Fig. 4f and Supplementary Fig. 4a, b). Moreover, *FZD7* depletion impaired invasion through 3D collagen I matrix (Fig. 4g). *FZD7* effects were further validated using two different shRNAs (Supplementary Fig. 4c–g). These data suggest that

**Fig. 2 EMT genes regulated by ROCK1/2 control amoeboid invasive features. a** Heatmap displaying enrichment scores for differentially expressed EMT and metastasis-related signatures in amoeboid A375M2 cells compared to A375P cells or to A375M2 cells treated with ROCKi (H1152 and Y27632) or blebbistatin using ssGSEA. **b** Representative phase-contrast images of A375M2 cells on top of collagen I matrix (left) and immunoblots of p-MLC2 in A375M2 and WM1361 cells (right) after 4 h treatment with ROCKi (H1152) ($n = 3$). Scale bar, 50 μm. **c** Fold regulation of EMT-related gene expression from EMT-directed qPCR array in A375M2 and WM1361 cells treated with ROCKi (H1152) for 4 h ($n = 4$). **d** Representative confocal images of p-MLC2 and F-actin staining in WM1361 cells on collagen I matrix after *WNT11* knockdown ($n = 3$). Scale bar, 20 μm. **e** Quantification of cell morphology (>280 cells pooled from $n = 3$) and **f** p-MLC2 immunofluorescence signal normalized by cell area (>85 cells pooled from $n = 3$) in WM1361 cells on collagen I matrix after depletion of indicated genes. **g** Representative confocal images (left) and quantification (right) of adhesion of WM1361 cells to a monolayer of keratinocytes after depletion of indicated genes ($n = 5$ for *SERPINE1*, $n = 4$ for *TCF4*, $n = 3$ for *WNT11*, *WNT5B*, *AHNAK* and *CAV2*). Scale bar, 20 μm. **h** Representative confocal images (left) and quantification (right) of 3D invasion index through a collagen I matrix of WM1361 cells after depletion of indicated genes ($n = 3$). Scale bar, 50 μm. **i** Summary table of amoeboid functional assays on WM1361 cells after *SERPINE1*, *WNT11*, *WNT5B*, *AHNAK*, *TCF4* and *CAV2* knockdown. "+" sign indicates a significant phenotype of the corresponding gene knockdown. **c**, **g**, **h** Graphs show mean ± s.e.m. **e**, **f** Box limits show 25th and 75th percentiles, the horizontal line shows the median, and whiskers show minimum and maximum range of values. **b–h** $n$ means number of independent biological experiments. **c** Two-tailed $t$-test with Benjamini, Krieger and Yekutieli correction for multiple comparisons. **e**, **f** Kruskal–Wallis test with Benjamini, Krieger and Yekutieli correction. **g**, **h** One-way ANOVA with Benjamini, Krieger and Yekutieli correction. For all graphs, *$p < 0.05$, **$p < 0.01$, ***$p < 0.001$, ****$p < 0.0001$. The exact significant $p$ values for *$p$, **$p$ and ***$p$ are provided in Supplementary Table 1.

FZD7 is an important regulator of amoeboid cell invasion and melanosphere formation.

After binding to their cognate receptors, non-canonical Wnt ligands can signal through two key molecular platforms: Dvl-PLCB mediated calcium signalling[36] and Dvl-DAAM1-Rho signalling[37,38]. To establish specificity, we investigated the effects of *PLCB1* and *DAAM1* depletion. *PLCB1* was chosen among its different isoforms as it is upregulated in melanoma metastasis (Supplementary Fig. 4h). Either *PLCB1* or *DAAM1* silencing decreased basal levels of cell rounding and p-MLC2 (Fig. 4h and Supplementary Fig. 4i, j), indicating that both pathways are able to control basal Myosin II activity. However, only *DAAM1* but not *PCLB1* depletion impaired melanosphere formation in both A375M2 and WM1361 cells (Fig. 4i) without affecting 2D cell viability (Fig. 4j). In addition, *DAAM1* silencing decreased invasion through 3D collagen I matrix (Fig. 4k). *DAAM1* effects were further validated using two different shRNAs (Supplementary Fig. 4k–o). Moreover, expression of stem cell-related markers was decreased in sh*DAAM1* cells (Supplementary Fig. 4p). These results suggest an important role of DAAM1 controlling amoeboid invasion and tumour-initiating potential.

We have shown that WNT11 can promote amoeboid features in melanoma cells in a paracrine manner (Fig. 3k). Crucially, we found that the increase in cell rounding measured after WNT11 stimulation was no longer observed in *DAAM1* depleted cells, while *PLCB1* depleted cells still responded to the ligand (Fig. 4l and Supplementary Fig. 4q). Furthermore, WNT11 stimulated RhoA activity and downstream Myosin II levels in melanoma cells, but this effect was lost when *DAAM1* was silenced (Fig. 4m, n).

Altogether, these results show that WNT11/5B-FZD7-DAAM1 pathway control melanosphere formation and amoeboid invasive behaviour by activation of a specific pool of RhoA-ROCK1/2-Myosin II.

**FZD7-DAAM1-RhoA-ROCK1/2 supports tumour initiation and metastasis in vivo.** We next tested whether the FZD7-DAAM1-RhoA-ROCK1/2-Myosin II axis might affect melanoma in vivo. First, A375M2 cells were pre-treated with different ROCKi (H1152 and GSK269962A) ex vivo for 5 days. Relative number of cells were reduced (Supplementary Fig. 5a). Nevertheless, only viable cells were subsequently injected subcutaneously into immunodeficient NSG mice in equal numbers, without any further treatment of animals. 18 days post-injection, ROCKi pre-treated A375M2-derived tumours showed over 60% reduction in tumour weight (Fig. 5a). Importantly, even if these

tumours had never received any treatment in vivo, we measured reduced cell rounding, p-MLC2 levels and ki-67 positive cells in ROCKi pre-treated tumours (Fig. 5b–d). Similarly, ROCKi pre-treated B16F10 melanoma cells (Supplementary Fig. 5b, c) injected subcutaneously into immunocompetent C57BL/6J mice showed reduced tumour growth (Supplementary Fig. 5d) and decreased amoeboid cell number in ROCKi pre-treated group (Supplementary Fig. 5e–g). These results show that ROCK1/2 supports the intrinsic abilities of both human and mouse melanoma cells to form tumours by supporting amoeboid features.

To examine the functional role FZD7 and DAAM1 might play in tumour initiation in vivo, limiting dilutions of either sh*FZD7* or sh*DAAM1* WM1361 cells were injected subcutaneously into NSG mice. Reduced *FZD7* levels (Supplementary Fig. 5h) led to a decrease in tumour growth when 50,000 and 5,000 cells were injected, while no changes were observed with a saturating concentration of 500,000 cells (Fig. 5e). Importantly, there was a significant reduction in TIF using ELDA (Fig. 5e). An enrichment in rounded cells (Fig. 5f) and p-MLC2 levels (Fig. 5g and Supplementary 5i) was observed at the IF of shControl tumours compared with TBs, while loss of cell rounding and reduced Myosin II activity were measured in sh*FZD7* derived tumours (Fig. 5f, g). sh*FZD7* tumours showed a decrease in cells with very high Myosin II in the IF (Supplementary Fig. 5i). Although there were no differences in cell numbers in vitro, sh*FZD7* tumours in vivo displayed a reduction of the fraction of cells positive for ki-67 proliferative marker (Fig. 5h). These data demonstrate that FZD7 loss results in a reduction of the amoeboid proliferative cell fraction in the tumour, leading to reduced tumour initiation.

We next assayed the in vivo tumour-initiating capacity of limiting dilutions of sh*DAAM1* WM1361 cells. *DAAM1* depletion (Supplementary Fig. 5j) yielded a reduction in TIF and a decrease in tumour growth at limiting dilutions of 50,000 and 5,000 cells (Fig. 5i). sh*DAAM1* tumours displayed pronounced loss of cell rounding (Fig. 5j), Myosin II activity (Fig. 5k and Supplementary Fig. 5k) and ki-67 positive cells (Fig. 5l). The in vivo tumour-initiating capacity of DAAM1 in melanoma was further validated using an additional melanoma cell line A375M2 (Supplementary Fig. 5l–p). *DAAM1* depletion led to a reduction in TIF and in A375M2 tumour growth (Supplementary Fig. 5l). These results confirm that FZD7-DAAM1-RhoA-ROCK1/2 promote tumour initiation by sustaining melanoma amoeboid behaviour.

Cells at the IF of tumours are those strategically positioned to leave the primary tumour and disseminate to distant sites. The lung is one of the main sites to which melanoma metastasizes and high Myosin II levels in amoeboid cells promote lung seeding and

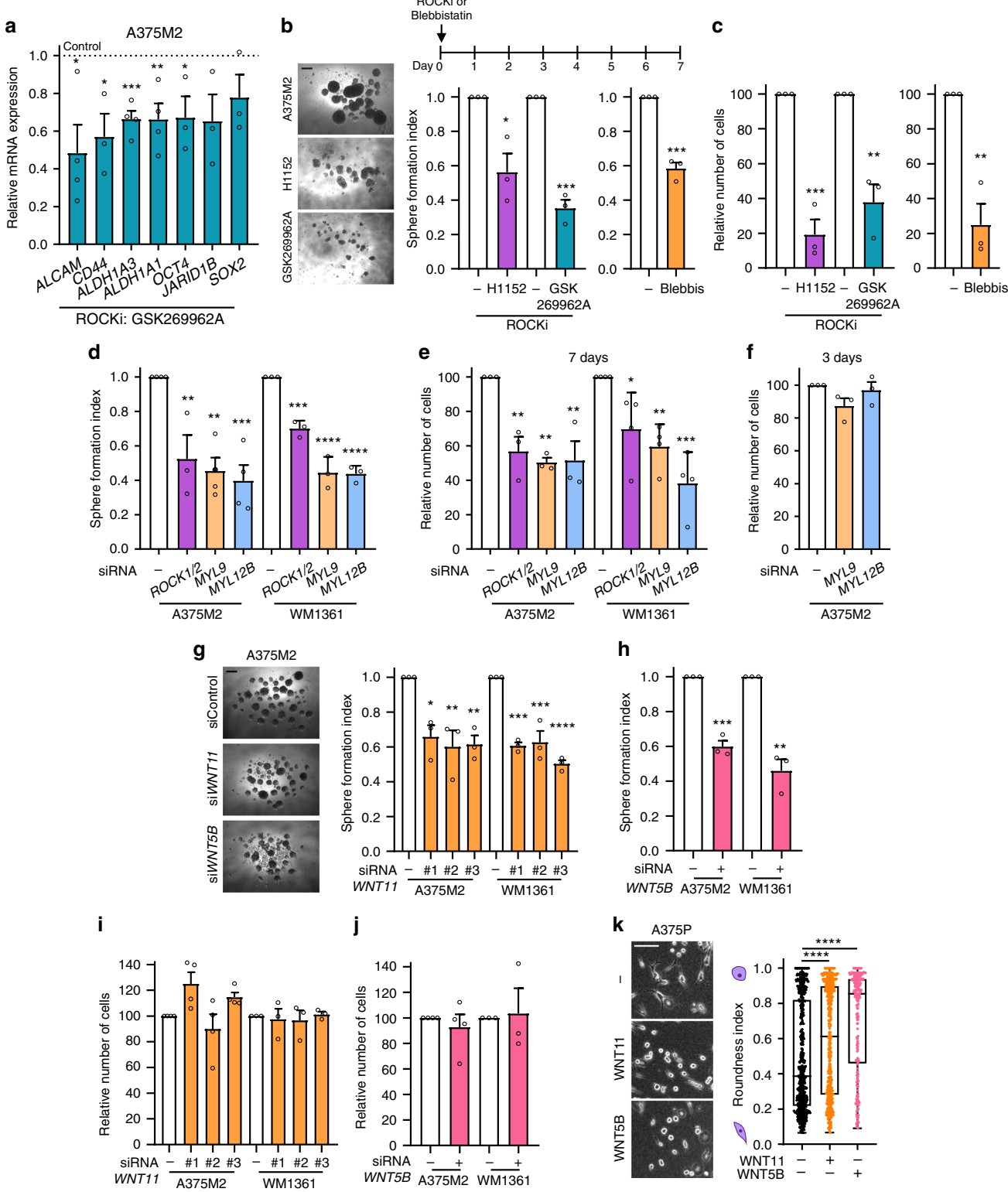

colonization[7,12,14]. To test the role of DAAM1 in both processes, we used an experimental metastasis model. Similar cell numbers lodged in the lung vasculature 30 min post-tail vein injection (Supplementary Fig. 5q). However, sh*DAAM1* melanoma cells were less efficient in colonising the lung after 24 h (Fig. 5m). Strikingly, the area of metastatic lesions 3 weeks after was reduced when *DAAM1* was depleted (Fig. 5m). These results indicate that DAAM1 is important for early stages of lung colonization and for the ability to outgrow at later stages. Our data further suggest that loss of amoeboid features has an impact both in early lung seeding and later in metastatic outgrowth.

**Amoeboid behaviour enhances tumour formation, tumour progression and metastasis in vivo.** We have shown that the ROCK1/2-Myosin II amoeboid phenotype sustained by FZD7-DAAM1 signalling supports tumour formation and dissemination.

**Fig. 3 Non-canonical Wnt ligands support melanosphere formation and amoeboid behaviour. a** mRNA expression of stem cell-related markers by qRT-PCR in A375M2 cells treated with ROCKi (GSK269962A) for 24 h compared to control A375M2 cells ($n = 4$ for *ALCAM, ALDH1A3, ALDH1A1*; $n = 3$ for *CD44, OCT4, JARID1B, SOX2*). **b** Schematic of treatment (top), representative phase-contrast images (bottom left) and quantification of sphere formation index (bottom right) and **c** cell viability of A375M2 cells and A375M2 cells treated with one dose of ROCKi (H1152 or GSK269962A) or blebbistatin ($n = 3$). Scale bar, 250 µm. **d–f** After *ROCK1/2, MYL9* or *MYL12B* knockdown in A375M2 and WM1361 cells, quantification of **d** sphere formation index ($n = 4$ for A375M2, $n = 3$ for WM1361) and **e, f** cell viability after **e** 7 days ($n = 3$ for A375M2, $n = 4$ for WM1361) or **f** 3 days ($n = 3$) of cell seeding. **g–j** After *WNT11* or *WNT5B* knockdown in A375M2 and WM1361 cells, **g, h** representative phase-contrast images (left) and quantification of sphere formation index (right) ($n = 3$) and **i, j** cell viability ($n = 4$ for A375M2, $n = 3$ for WM1361). Scale bar, 250 µm. **k** Representative phase-contrast images (left) and quantification of cell morphology (right) of A375P cells after 24 h of WNT11 or WNT5B stimulation (>225 cells pooled from $n = 3$). Scale bar, 100 µm. **a–j** Graphs show mean ± s.e.m. **k** Box limits show 25th and 75th percentiles, the horizontal line shows the median, and whiskers show minimum and maximum range of values. **a–k** $n$ means number of independent biological experiments. **a–c, h, j** Two-tailed *t*-test. **d–g, i** One-way ANOVA with Dunnett post-hoc test. **k** Kruskal–Wallis with Dunn's multiple comparison test. For all graphs, $*p < 0.05$, $**p < 0.01$, $***p < 0.001$, $****p < 0.0001$. The exact significant $p$ values for $*p$, $**p$ and $***p$ are provided in Supplementary Table 1.

Using an orthotopic spontaneous melanoma metastasis model, we next recapitulated the steps of melanoma progression. Injection of 4599-BRAF[V600E] melanoma cells into the dermis of NSG mice resulted in the formation of orthotopic tumours. Mice were then systemically treated with ROCKi (Fig. 6a) since to our knowledge no drugs were available for the effective/specific inhibition of non-canonical FZD7-DAAM1 signalling in vivo. We observed three defined areas in control tumours: TB, IF and a further area of local invasion into the dermis (invading cells or distal invasive front (DIF)) (Fig. 6b) closely recapitulating the human disease[39]. All invading cells at the DIF displayed rounded-amoeboid features: 56% of them exhibited the highest Myosin II activity and 60.4% of invading cells were positive for the proliferative marker ki-67 (Fig. 6c). Specifically, co-staining of p-MLC2 score 3 and ki-67 was found in 30.2% of invading cells at the DIFs (Fig. 6c). These data suggest that amoeboid invading cells are also proliferative. Cell rounding (Fig. 6d), overall Myosin II activity (Fig. 6e) and ki-67 positive cells (Fig. 6f) were progressively increased from TB to IF to DIF in control tumours. Cells with very high Myosin II were abundant in the IF and further enriched in the DIF (Fig. 6e and Supplementary Fig. 6a, b). Importantly, ROCKi led to loss of cell rounding and decreased Myosin II activity at IFs and DIFs (Fig. 6d, e), while a reduction of ki-67 positive cells was observed in all areas of ROCKi treated tumours (Fig. 6f). These data suggest that ROCKi impairs amoeboid features and proliferation in vivo.

Phenotype switching from proliferative to invasive states has been implicated in melanoma progression[18–20]. However, our transcriptional analysis showed that amoeboid A375M2 cells were enriched in both proliferative and invasive genes (Fig. 6g and Supplementary Fig. 6c, d). Moreover, enrichment in Myosin II activity and ki-67 positive cells was found in IFs of A375M2 tumours (Fig. 1e and Supplementary Fig. 1a, c), suggesting that the amoeboid phenotype sustained by ROCK1/2-Myosin II could, in principle, support tumour formation and tumour dissemination. Remarkably, in accordance with its role in regulating both proliferative and invasive genes, ROCKi caused a reduction in primary tumour growth in 4599 tumours (Fig. 6h) and resulted in loss of melanoma cells leaving the primary tumour and invading into adjacent tissue (Fig. 6i, j). In line with reduced local invasion, spontaneous lung metastasis was reduced dramatically in ROCKi treated mice (Fig. 6k). Our results suggest that ROCK1/2-Myosin II activity in amoeboid melanoma cells controls both tumour growth and spread.

In the previous experimental setting, as both primary tumour growth and local invasion were reduced, it was difficult to separate these effects from those on metastasis. We next evaluated the impact of ROCKi in melanoma metastatic colonisation and outgrowth using tail vein injection assays. 4599 cells were pre-treated with ROCKi for 24 h prior to intravenous injection and continued treatment in vivo for 6 days. Cell survival in the blood stream was not altered as similar cell numbers lodged in the lungs 30 min post-tail vein injection (Supplementary Fig. 6e). However, ROCKi reduced lung metastatic establishment (Fig. 6l). Reduced lung metastasis was also measured when 4599 cells -without any pre-treatment- were intravenously injected, while 4 h after injection mice received ROCKi systemic treatment for 12 days (Fig. 6m).

Our data confirm that ROCK1/2-Myosin II activity is not only important for tumour formation and invasion but also for early colonization and later metastatic growth. This suggests that ROCK1/2-Myosin II activity plays a crucial role in all steps of melanoma progression.

**Analysis of the invasive front of human primary melanomas.** To understand if amoeboid cells in the IF have increased tumour-initiating features in the human clinical setting, we analysed matched TBs and IFs of 53 human primary melanomas (Supplementary Table 2). Cell morphology and p-MLC2 levels were assessed as amoeboid markers; ki-67 staining as a proliferative marker; WNT11, WNT5B and DAAM1 expression as non-canonical Wnt markers; and ALDH1A1, ALDH1A3, CD44, NANOG and OCT4 were evaluated as cancer stem cell-related markers. In accordance with previous assessments in melanoma patients[7,10,12,15], we could measure that IFs of human melanomas were enriched in rounded-amoeboid cells with high Myosin II activity (Fig. 7a, b and Supplementary Fig. 7a). Importantly, as observed in our tumour models (Figs. 1, 5, 6), cells with very high Myosin II were predominant at human IFs. Moreover, an increase in the percentage of ki-67 positive cells was also found at IFs (Fig. 7c). Interestingly, cells with higher levels of DAAM1, WNT11 and WNT5B were also enriched in the IFs (Fig. 7d–f). Cells with score 3 level of DAAM1 were predominant in these areas (Supplementary Fig. 7b). Importantly, we found an enrichment in ALDH1A1, CD44 and NANOG expression at these IFs (Fig. 7g–i), while ALDH1A3 and OCT4 did not change (Supplementary Fig. 7c, d). We provided regional information showing high expression of amoeboid/non-canonical Wnt/stem cell markers in the IF area of human primary tumours (Supplementary Fig. 7e). Interestingly, principal component analysis of all differentially expressed markers clearly separated TB and IF areas in primary melanomas (Fig. 7j). Amongst all markers investigated, ALDH1A1 enrichment was highest, showing a median 13-fold increase in IFs. Strikingly, high ALDH1A1 protein expression in the IF appeared to confer worse prognosis and shorter disease-free survival in our cohort of melanoma patients (Fig. 7k, l). ALDH1A1 is a functional marker involved in self-renewal and proliferation with key anti-oxidative roles[40]. Invasive melanoma cells with high Myosin II harbour lower levels of reactive oxygen species (ROS)[8,24], while antioxidants promote

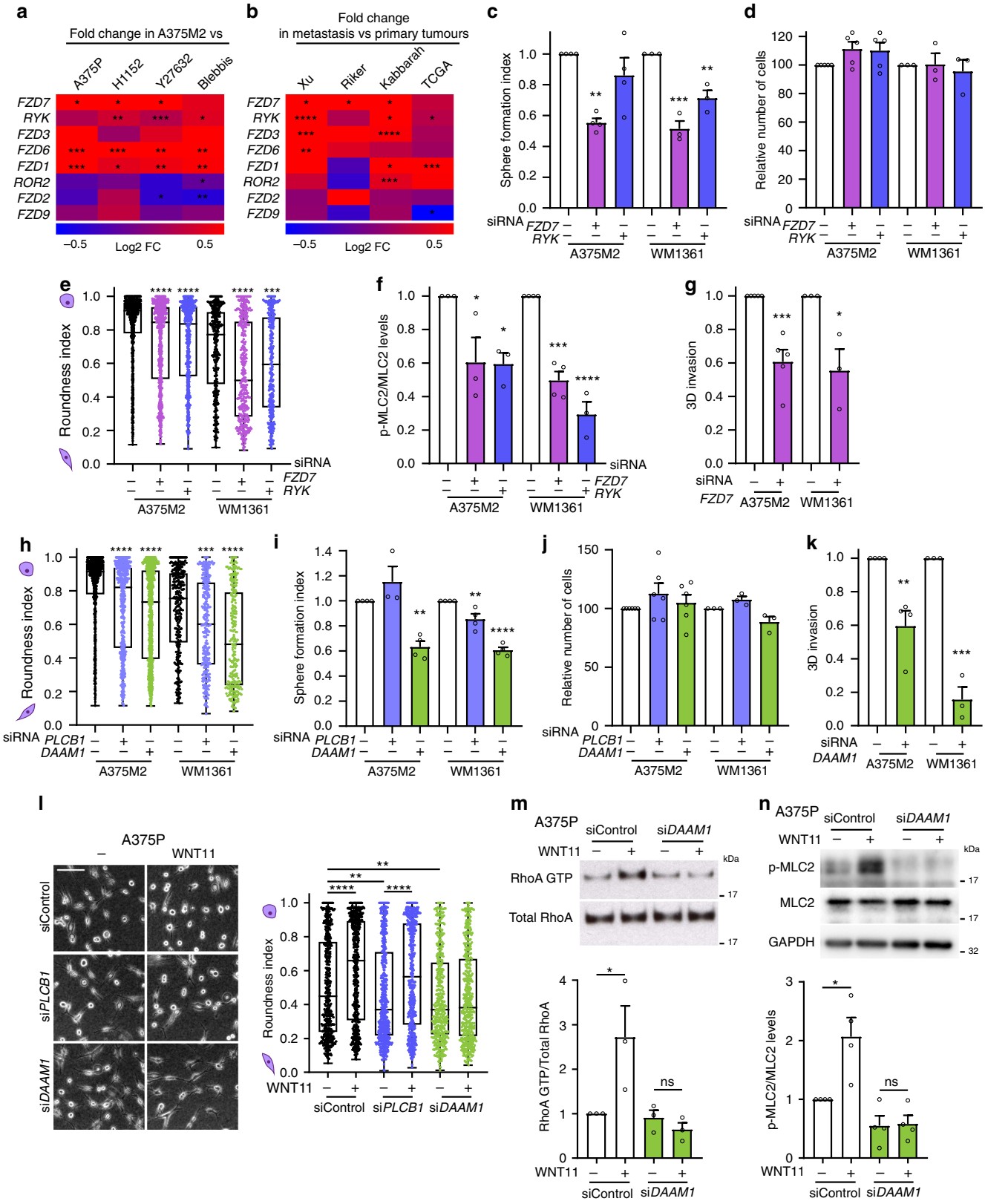

melanoma invasion and metastasis[41,42]. Moreover, higher ROS levels reduce Myosin II activity[8]. Silencing *ALDH1A1* led to decreased melanosphere formation (Supplementary Fig. 7f, g). Interestingly, *ALDH1A1* depletion resulted in loss of cell rounding and lower p-MLC2 levels (Supplementary Fig. 7h, i), while ROS levels, as measured by FACS, were increased (Supplementary Fig. 7j, k). Our data suggest that ALDH1A1-supports amoeboid tumour-initiating features via regulation of ROS metabolism. Overall, these data illustrate that the IFs of human primary melanomas are tumour areas enriched in amoeboid, proliferative, non-canonical Wnt and cancer stem cell-related markers.

**Fig. 4 FZD7 downstream of WNT11 supports melanosphere formation and amoeboid invasion via DAAM1. a, b** Heatmap representing log2 fold change in expression of non-canonical Wnt receptors in **a** A375M2 cells compared to A375P cells or A375M2 cells treated with ROCKi (H1152 and Y27632) and blebbistatin and in **b** metastatic *versus* primary melanoma samples from indicated studies from GEO and TCGA databases. **c–f** After *FZD7* or *RYK* knockdown in A375M2 and WM1361 cells, quantification of **c** sphere formation index ($n = 4$ for A375M2, $n = 3$ for WM1361), **d** cell viability ($n = 5$ for A375M2, $n = 3$ for WM1361), **e** cell morphology (>300 cells pooled from $n = 3$) and **f** p-MLC2 levels ($n = 3$ for A375M2, $n = 4$ for WM1361). **g** Quantification of 3D invasion index through a collagen I matrix after *FZD7* knockdown in A375M2 ($n = 5$) and WM1361 cells ($n = 3$). **h–j** After *PLCB1* or *DAAM1* knockdown in A375M2 and WM1361 cells, quantification of **h** cell morphology (>250 cells pooled from $n = 3$), **i** sphere formation index ($n = 4$) and **j** cell viability ($n = 6$ for A375M2, $n = 3$ for WM1361). **k** Quantification of 3D invasion index through a collagen I matrix after *DAAM1* knockdown in A375M2 ($n = 4$) and WM1361 cells ($n = 3$). **l** Representative phase-contrast images (left) and quantification of cell morphology (right) of A375P cells on collagen I matrix after WNT11 stimulation and *PLCB1* or *DAAM1* knockdown (>400 cells pooled from $n = 5$). Scale bar, 100 μm. **m, n** Representative immunoblots (top) and quantification (bottom) of **m** RhoA-GTP in pulldown samples and total RhoA in total lysate ($n = 3$) and **n** p-MLC2 levels ($n = 4$) in A375P cells after WNT11 stimulation and *DAAM1* knockdown. **c, d, f, g, i–k, m, n** Graphs show mean ± s.e.m. **e, h, l** Box limits show 25th and 75th percentiles, the horizontal line shows the median, and whiskers show minimum and maximum range of values. **c–n** *n* means number of independent biological experiments. **a, g, k** Two-tailed *t*-test. **b** Two-tailed *t*-test with Welch's correction. **c, d, f, j** One-way ANOVA with Dunnett post-hoc test. **e, h** Kruskal–Wallis with Dunn's multiple comparison test. **l** Kruskal–Wallis test with Benjamini, Krieger and Yekutieli correction. **m, n** One-way ANOVA with Tukey post-hoc test. For all graphs, *$p < 0.05$, **$p < 0.01$, ***$p < 0.001$, ****$p < 0.0001$. The exact significant *p* values for *$p$, **$p$ and ***$p$ are provided in Supplementary Table 1.

---

**Analysis of the invasive front of human metastatic melanomas.** To further investigate if melanoma cells retain this phenotype during the course of secondary tumour formation, 45 matched TBs and IFs from human melanoma metastases were assessed (Supplementary Table 3). Metastatic melanomas retained the same regional distribution identified in primary tumours; that is, amoeboid (Fig. 8a, b and Supplementary Fig. 8a) and proliferative cells (Fig. 8c) were prominent in the IF areas, and that IFs also had an enrichment in non-canonical Wnt markers (Fig. 8d–f and Supplementary Fig. 8b) and stem cell-related markers (Fig. 8g–i and Supplementary Fig. 8c, d). Importantly, the overall levels of all markers were higher in metastasis. In fact, TBs from metastatic lesions were more amoeboid, more proliferative and had a higher basal level of some non-canonical Wnt and stem cell-related markers (Supplementary Fig. 8e–h) when compared to primary tumour TBs. As a result, the increased expression of all markers in metastatic IFs was not as pronounced as in the primary tumour setting. These results show that the amoeboid tumour-initiating phenotype found in primary IFs is recapitulated and further enhanced in metastatic melanoma specimens. Overall, our data show that amoeboid cell invasive/proliferative behaviour is selected during melanoma progression.

## Discussion

Melanoma is a highly aggressive tumour that is able to metastasize from very early stages of the disease. The high migratory ability of melanoma cells has been explained in part by its NC origin[1]. Signalling pathways involved in NC development and melanoma progression are closely related[2,3]. Here, we provide evidence that amoeboid melanoma cells display increased tumour-initiating abilities. We show that rounded-amoeboid cells with high Myosin II activity and high levels of ki-67 are associated with increased tumour initiation capacity in vitro and in vivo. ROCK1/2-Myosin II pathway is a key regulator of cell proliferation, migration, invasion and metastatic behaviour[7,9–11,29–31]. We have recently reported that Myosin II activity drives therapy resistance in melanoma[24], a mechanism closely related with tumour-initiating cell properties[25,43]. NC stem cell transcriptional states play a critical role in driving relapse and drug resistance in melanoma[4]. Moreover, analysis of our transcriptional signature for amoeboid melanoma cells[10] reveals that amoeboid cells are enriched in NC and invasive signatures associated with drug resistance[4,24]. On the other hand, tumour-initiating cells are supposed to represent a small cell population in a slow proliferative state[25]. However, it has been described that they can proliferate vigorously and

tumour-initiating cell frequencies up to 25% have been reported[6,44,45]. Ki-67 can affect clonogenic stem cell properties[46], and sub-populations of ki-67 proliferating stem cells have also been found in some tumours[26].

Importantly, we also find that amoeboid melanoma cells express EMT-related genes driven by ROCK1/2. Rho-ROCK1/2 signalling has been shown in other models to regulate actin cytoskeleton rearrangements during EMT[47] while promoting transcriptional changes through activating p38 signalling[48,49]. Interestingly, ROCK1/2 regulates TGFβ expression in our EMT array and TGFβ signalling is one of the top networks associated with the EMT gene signature regulated by ROCK1/2. We have also shown that amoeboid cells secrete high levels of TGFβ and its secretion is ROCK1/2 dependent[7,15,24]. TGFβ is a key driver of EMT by inducing transcription of several mesenchymal genes and increasing the activity of EMT transcription factors[47,50]. This suggests TGFβ as one of the possible mechanisms by which ROCKs modulate a global EMT gene program and, on the other hand, amoeboid invasion[7].

Among these EMT-related genes regulated by ROCKs, we find Wnt ligands. Such genes, in turn, drive amoeboid features. Specifically, we find that the WNT11/5B-FZD7-DAAM1 signalling axis has an impact on tumour initiation by controlling the amoeboid phenotype. WNT11-DAAM1 signalling has been described to play an essential role in NC migration[51–53], NC formation and NC specification during early embryonic development[54,55]. WNT5A through $Ca^{2+}$ and PKC signalling has been described to play a role in melanoma progression[56], while other Wnt ligands have not been studied in detail. We show that WNT11 and WNT5B control amoeboid self-renewal and invasive behaviour. Moreover, melanoma cells could exploit paracrine Wnt signalling from stroma[32], sustaining the amoeboid phenotype, as long as they express the correct receptor repertoire. We find that FZD7 receptor downstream of WNT11/WNT5B ligands is upregulated during melanoma progression and we show this receptor sustains amoeboid behaviour. Limiting-dilution analysis revealed how FZD7 supports in vivo tumour-initiating abilities and tumour growth in melanoma, consistent with previous reports in other tumours[57]. Importantly, our work has unravelled a crucial role for DAAM1 in promoting both tumour and metastasis initiation and later metastatic outgrowth in melanoma by regulating amoeboid features. DAAM1 is a formin-homology protein that plays a central role in translating signals from Wnt ligands to the actin cytoskeleton[37]. Unlike other formins, DAAM1 activation is specifically mediated by its binding with Dvl upon Wnt stimulation rather than by its interaction with

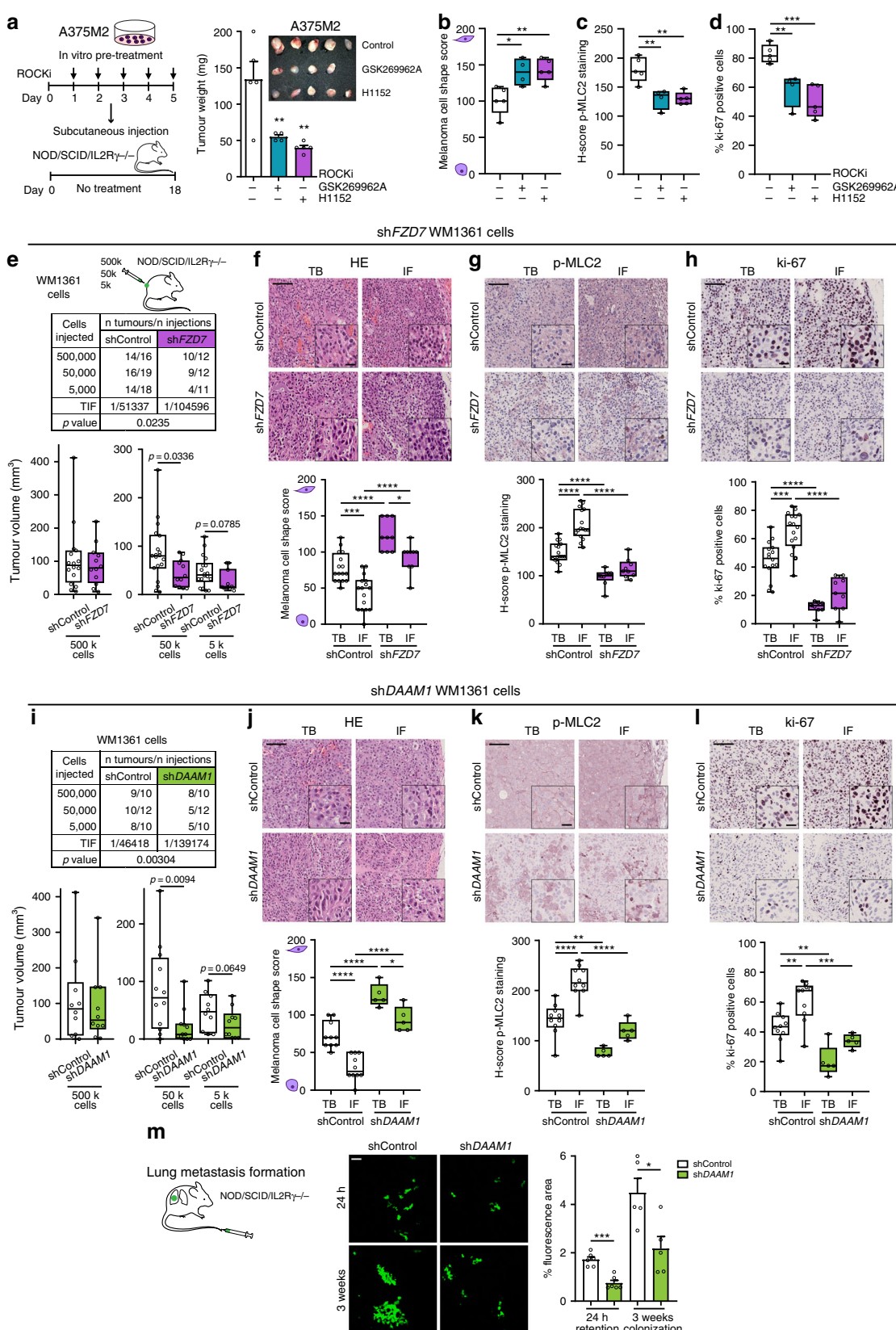

active Rho GTPases[37,38]. Such mechanisms have been described during development, but not during cancer initiation or progression. Therefore, we highlight here yet another example of cancer cells hijacking a developmental program. On the other hand, DAAM1 promotes RhoA activation[37,38] via recruitment of PDZ-RhoGEF[58]. In accordance, using the Human Protein Atlas,

we find that this PDZ-RhoGEF is highly expressed in melanoma compared to other cancers (Supplementary Fig. 8i). We propose here that non-canonical WNT11/5B-FZD7-DAAM1 axis crucially and specifically controls tumour initiation potential in melanoma by promoting ROCK1/2-Myosin II driven amoeboid proliferative and invasive phenotype (Fig. 8j).

**Fig. 5 FZD7-DAAM1-RhoA-ROCK1/2 supports tumour initiation and metastasis in vivo. a** Schematic of experiment (left) and tumour weight (right) of ROCKi (H1152 or GSK269962A) pre-treated A375M2 cells 18 days post-subcutaneous injection into NSG mice ($n = 5$ mice for control and H1152, $n = 4$ for GSK269962A). **b–d** Quantification of **b** melanoma cell shape score, **c** H-score of p-MLC2 staining and **d** ki-67 positive cells in A375M2 tumours from (**a**). **e** Limiting dilution assay estimating TIF (top) and tumour volume (bottom) of shControl and sh*FZD7* WM1361 cells when injected at different dilutions (500,000, 50,000 and 5,000 cells) into NSG mice (Number of tumours per condition indicated in table). TIF was determined using ELDA. **f–h** Representative images (top) and quantification (bottom) of **f** melanoma cell shape score, **g** H-score of p-MLC2 staining and **h** ki-67 positive cells in TB and IF of shControl ($n = 16$) and sh*FZD7* ($n = 9$) derived tumours from 50,000 cells' condition from (**e**). Scale bar, 100 μm; inset, 25 μm. **i** Limiting dilution assay estimating TIF (top) and tumour volume (bottom) of shControl and sh*DAAM1* WM1361 cells when injected at different dilutions (500,000, 50,000 and 5,000 cells) into NSG mice (Number of tumours per condition indicated in table). TIF was determined using ELDA. **j–l** Representative images (top) and quantification (bottom) of **j** melanoma cell shape score, **k** H-score of p-MLC2 staining and **l** ki-67 positive cells in TB and IF of shControl ($n = 10$) and sh*DAAM1* ($n = 5$) derived tumours from 50,000 cells' condition from (**i**). Scale bar, 100 μm; inset, 25 μm. **m** Representative confocal images (left) and percentage of fluorescence area (right) of mouse lungs 24 h ($n = 6$ mice) and 3 weeks ($n = 5$ mice) after tail vein injection into NSG mice of WM1361 cells expressing shControl and sh*DAAM1*. Scale bar, 50 μm. **a, m** Graphs show mean ± s.e.m. **b–l** Box limits show 25th and 75th percentiles, the horizontal line shows the median, and whiskers show minimum and maximum range of values. **a–d** One-way ANOVA with Dunnett post-hoc test. **e, i** Two-tailed Mann–Whitney test. **f–h, j–l** One-way ANOVA with Tukey post-hoc test. **m** Two-tailed *t*-test. For all graphs, *$p < 0.05$, **$p < 0.01$, ***$p < 0.001$, ****$p < 0.0001$. The exact significant p values for *p, **p and ***p are provided in Supplementary Table 1. Schematics in this figure were created using Servier Medical Art templates licensed under a Creative Commons Attribution 3.0 Unported License (https://smart.servier.com).

Furthermore, we provide evidence that amoeboid cells are selected during several steps of melanoma progression. In the current study, we use potent and selective ROCKi to demonstrate that ROCK1/2-Myosin II axis is an important regulator of tumour growth and metastasis in melanoma. We have reported that amoeboid melanoma cells support tumour growth by reprogramming the immune microenvironment[15]. Furthermore, the tumour microenvironment can regulate ROCK2-Myosin II to support tumour growth[59]. We demonstrate here that in melanoma ROCK1/2-Myosin II also plays an important pro-tumourigenic role via cell autonomous mechanisms. Importantly, amoeboid melanoma cells express genes from both proliferative and invasive signatures[18,19]. We show the in vivo existence of amoeboid invading cells that are positive for the ki-67 proliferative marker and harbour very high levels of Myosin II. Such a population is also impaired when blocking the whole pathway with ROCKi. In a recent study, a subpopulation of melanoma cells has been identified that simultaneously displays proliferative and invasive properties as a result of high transcriptional TGFβ signalling activation[60]. Since their gene ontology analysis show an enrichment in amoeboid features[60], it is tempting to speculate that this is the same population of melanoma cells.

In this study, we demonstrate that the amoeboid proliferative and invasive phenotype promoted by non-canonical WNT-DAAM1 signalling is required for tumour initiation in melanoma and for both initial and late metastatic stages. We suggest that amoeboid melanoma cells sustain the expression of an EMT and stem cell-related gene signature to successfully metastasize. In agreement with this concept, assessment of patient biopsies revealed that IFs of human primary melanomas are enriched in amoeboid, proliferative, non-canonical Wnt and cancer stem cell-related markers. Functional stem cells that drive tumour expansion and proliferative cells have been described to reside at the tumour edge of some tumours[61]. Moreover, ki-67 index and mitotic rate are indicative of neoplastic progression and appear as prognostic factors in melanoma[23,62,63]. Furthermore, within this amoeboid tumour-initiating phenotype, we identify ALDH1A1, a key gene for tumour-initiating abilities. Importantly, loss of ALDH1A1 results in loss of the amoeboid phenotype and vice versa. Since ALDH1A1 has been implicated in tumour-initiating capacity in melanoma and has been associated with drug resistance[64,65], we propose ALDH1A1 as a potential prognostic marker in melanoma patients. Importantly, we find metastatic melanoma lesions to recapitulate the behaviour of primary tumours and show an enhancement of the amoeboid tumour-initiating pattern.

Overall, we have unveiled how WNT11B/5B-FZD7-DAAM1 in amoeboid cells supports tumour-initiating properties, while also promoting invasion. Since amoeboid cells are prominent at the edge of tumours, our work sheds some light into the early ability of melanomas to disseminate and grow at distant sites. We propose that the IFs of melanoma tumours are areas with important prognostic features that pathologists should carefully evaluate. Furthermore, after surgical melanoma removal, we suggest inhibiting non-canonical Wnt signalling to eradicate aggressive amoeboid behaviour at the edge of tumours.

## Methods

**Cell culture.** Cells were grown at 37 °C and 10% $CO_2$ in DMEM (A375M2, A375P, B16F10, 4599, HEK293T and HaCaT) or RPMI (WM1361, WM983B and WM983A) supplemented with 10% FBS and 1% penicillin/streptomycin (all from Gibco). A375M2 and A375P cells were from Prof. Richard Hynes (HHMI, MIT, USA), WM1361 cells were from Prof. Richard Marais (Cancer Research UK Manchester Institute, UK), WM983B and WM983A were purchased from Coriell Institute, B16F10 cells were from Dr. Hector Peinado (CNIO, Spain), 4599 cells were from Dr. Amine Sadok (Institute of Cancer Research, UK) and Prof. Richard Marais, HEK293T cells were from Dr. Jeremy Carlton (The Francis Crick Institute, UK), and HaCaT cells were from Dr. Ester Martin-Villar (IIBM, UAM, Spain). A375M2, A375P, WM983B, WM983A and 4599 were authenticated using short tandem repeat DNA profiling. All cell lines were routinely tested for mycoplasma contamination. All cell lines were kept in culture for a maximum of three to four passages and cell phenotypes were verified routinely.

**Sphere formation assay.** The sphere-forming assay was performed as previously described[66]. Briefly, individual cells were plated in 100 μl of Sphere Media (DMEM/F12 supplemented with 1:50 B27 (Invitrogen), 20 ng/ml EGF (PHG0315, Invitrogen), 20 ng/ml FGF (PHG0026, Invitrogen), 4 mg/ml heparin (H3149, Sigma) and 1% penicillin/streptomycin) in ultra-low-attachment 96-well plates (Costar) at a density of 1,000 cells/well and kept at 37 °C and 10% $CO_2$. For serial passages, melanospheres were collected by centrifugation followed by enzymatic (trypsin treatment for 10 min at 37 °C) and mechanical dissociation using 25G needles. The resulting single-cell suspension was counted and re-plated at the original density. In all assays, melanosphere formation was quantified after 7 days. The Sphere Formation Index (SFI) considered both, number and size of the spheres, and was calculated by summing the area of all spheres formed, divided by the number of single cells initially plated. In each biological experiment, 2–3 technical replicates were performed and average SFI considered. Quantitative analysis of images was performed using ImageJ.

**Cell culture on thick layers of collagen I.** Bovine collagen I (no. 5005-B; PureCol, Advanced BioMatrix) thick layer was prepared at 1.7 mg/ml as previously described[10,12]. After polymerization (4 h), cells were seeded on top of collagen in medium containing 10% FBS, allowed to adhere for 16 h and media was changed to 1% FBS with corresponding treatments (where appropriate). Cells were analysed 24 h later. At the end of the experiment, gels were fixed with 4% formaldehyde and imaged or lysates were collected.

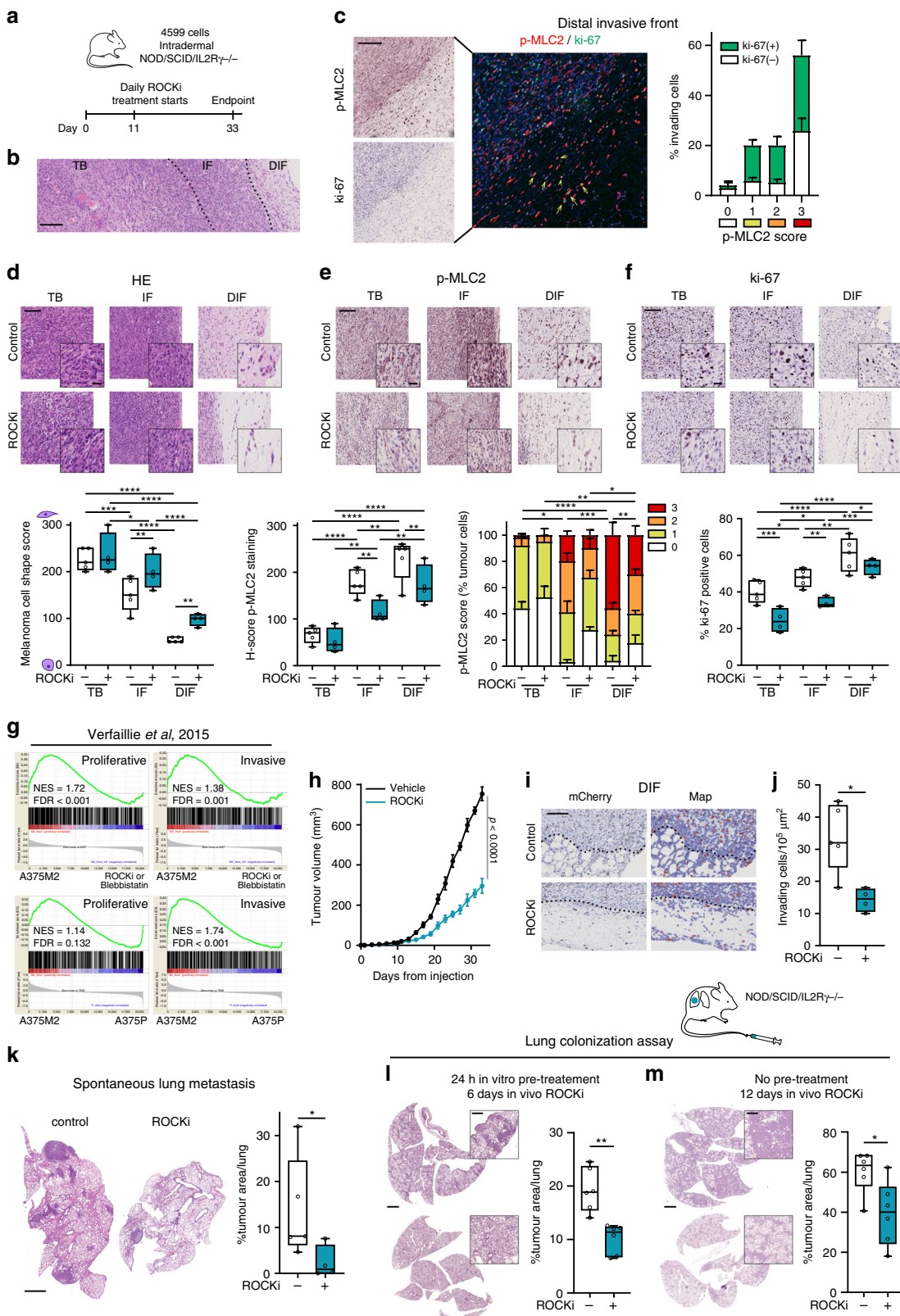

## Analysis of cell morphology

Cell morphology was quantified on phase-contrast images of cells cultured on top of bovine collagen I matrices using ImageJ as previously described[12,15,24]. Cell morphology was assessed using the morphology descriptor tool "roundness" after manually drawing around the cell shape. Roundness index is calculated as $4 \times area/\pi \times major\_axis\_length^2$. Values closer to 1 represent rounded morphology; values closer to 0 represent more spread and/or spindle-shaped cells.

## Drug treatments

Contractility inhibitors and concentrations used in this study were: 5 μM H1152 (stock resuspended in water; #555550, Calbiochem), 1 μM or 5 μM GSK269962A (stock resuspended in DMSO; #1167, Axon) and 25 μM blebbistatin (stock resuspended in 95% DMSO; #203390, Calbiochem). Time of treatment was as specified in each experiment. WNT11 (200 ng/ml; 6179-WN-010, R&D Systems,) and WNT5B (500 ng/ml; 7347-WN-025, R&D Systems) were added in serum-free media for 24 h.

**Fig. 6 Amoeboid behaviour enhances tumour formation, tumour progression and metastasis in vivo. a** Schematic of in vivo experiment with 4599 cells injected intradermally into NSG mice treated with ROCKi (25 mg/kg GSK269962A) or control (5% DMSO) ($n = 5$ mice for control and $n = 4$ for ROCKi). **b** Representative H&E images showing areas of TB, IF and local area of invasion into the dermis (distal invasive front (DIF)) of primary tumours derived from (**a**). Dashed lines represent the boundary between tumour areas. Scale bar, 200 μm. **c** Representative images (left) and quantification (right) of invading cells with co-staining of p-MLC2 (red) and ki-67 (green) at the DIF of control tumours from (**a**). Scale bar, 200 μm. Yellow arrows indicate ki-67 positive invading cells with p-MLC2 score 3 intensity. **d–f** Representative images (top) and quantification (bottom) of **d** melanoma cell shape score, **e** H-score and percentage of cells with score intensity 0–3 for p-MLC2 staining and **f** ki-67 positive cells in primary tumours from (**a**). Scale bar, 100 μm; inset, 25 μm. **g** GSEA plots showing enrichment of proliferative and invasive gene signatures from Verfaillie study[19] in A375M2 cells compared to A375M2 cells treated with ROCKi (H1152 and Y27632) and blebbistatin or to A375P cells[10]. NES, normalized enrichment score; FDR, false discovery rate. **h** Tumour growth curves of 4599 cells from (**a**). **i** Representative images (left) and QuPath mark-up (right) of mCherry staining and **j** quantification of invading cells into the dermis in primary tumours derived from (**a**). Dashed lines represent the boundary between IF and DIF. Scale bar, 50 μm. **k** Representative images (left) and quantification of tumour area (right) of spontaneous lung metastasis from (**a**). Scale bar, 1 mm. **l, m** Representative images of mouse lungs (left) and quantification of tumour area (right) at the indicated times after tail vein injection of 4599 cells **l** pre-treated with ROCKi for 24 h or **m** without any pre-treatment into NSG mice. Animals were systemically treated with ROCKi ($n = 6$ mice/group). Scale bar, 2 mm; inset, 500 μm. **c** Graphs show mean ± s.e.m. **d–f**, **j–m** Box limits show 25th and 75th percentiles, the horizontal line shows the median, and whiskers show minimum and maximum range of values. **d–f** One-way ANOVA with Benjamini, Krieger and Yekutieli correction. **h, j** Two-tailed t-test. **k–m** Two-tailed Mann–Whitney test. For all graphs, *$p < 0.05$, **$p < 0.01$, ***$p < 0.001$, ****$p < 0.0001$. The exact significant p values for *$p$, **$p$ and ***$p$ are provided in Supplementary Table 1. Schematics in this figure were created using Servier Medical Art templates licensed under a Creative Commons Attribution 3.0 Unported License (https://smart.servier.com).

---

**Transfection and RNAi.** $2 \times 10^5$ cells/well were seeded on 6-well plates and transfected the next day with 20 nM siGenome SmartPool (SP) or On-Targetplus (OT) siRNA oligonucleotides, using Optimem-I and Lipofectamine 2000 (Invitrogen). For sphere formation assays and cell viability assays, cells were re-transfected after 24 h. In all assays, 48 h after transfection, cells were harvested and equal numbers were re-seeded for subsequent analyses. All siRNA sequences were from Dharmacon (Lafayette, USA) and are listed in Supplementary Table 4. In all siRNA experiments, Non-Targeting siRNA was used as control.

**Lentivirus generation and infection of melanoma cells.** $4 \times 10^5$ HEK293T cells on 6-well plates were transfected with lentiviral vector (1 μg) along with packaging vectors (p-MD2.VSVg (0.4 μg) and pΔ8.91 (1 μg)) using Optimem-I and Lipofectamine 2000 (Invitrogen). Supernatants with lentiviruses were collected 48 h after transfection, spun down and filtered (0.45 μm). For lentiviral transduction, $2 \times 10^5$ melanoma cells/well were seeded in 6-well plates and infected with lentiviruses. Stable cells were selected with 1 μg/ml puromycin (Life Technologies). *FZD7* and *DAAM1* silencing was achieved using pGIPZ Lentiviral shRNA constructs expressing TurboGFP from Dharmacon (Lafayette, USA): sh*FZD7* #1 (V3LHS_368263 5′-AGAGCACGCTGAAGACGCC-3′), sh*FZD7* #2 (V3LHS_368264 5′-CGTGTTTCATGATGGTGCG-3′), sh*DAAM1* #1 (V3LHS_339679 5′-TTGTGATTGGTCTCTCTCT-3′), sh*DAAM1* #2 (V3LHS_339678 5′-TTAGATTGAGAACACTGGG-3′), Non-silencing shControl (5′-ATCTCGCTTGGGCGAGAGTAAG-3′). EGFP-fused rat wild-type MCL2 plasmid was obtained from Dr. Tohru Takaki and Prof Erik Sahai[67].

**Immunoblotting and antibodies.** Cells were lysed in Laemmli sample buffer, boiled for 5 min, sonicated for 15 s and spun down. Lysates were fractionated using 10% or 12% SDS-polyacrylamide gel electrophoresis and transferred onto PVDF filters (0.45 μm, Immobilon). ECL or Prime ECL detection Systems (GE Healthcare) with HRP-conjugated secondary antibodies (GE Healthcare) were used for detection. Bands were quantified using Image J. Levels of phospho-proteins were calculated after correction to total levels of the relevant protein. Antibodies: pThr18/Ser19-MLC2 (1:750, #3674), MLC2 (1:750, #3672) and RhoA (1:1000, #2117) from Cell Signalling Technology; GFP (1:10000, sc-8334) from Santa Cruz Biotechnology; GAPDH (1:10000, MAB374) from Millipore.

**RhoA-GTP pulldown assay.** RhoA-GTP pulldown assays were performed as described[8,14]. $2 \times 10^5$ cells/well were seeded on 6-well plates, serum starved and treated with WNT11 as indicated above. Cells were lysed in lysis buffer containing 50 mM Tris pH 7.4, 10% glycerol, 1% NP40, 5 mM MgCl₂, 100 mM NaCl, 1 mM DTT and EDTA free protease inhibitor and spun down. A small proportion of protein lysates were separated for determination of total RhoA levels. The remaining protein lysate was incubated with glutathione S-transferase (GST)-conjugated with Rhotekin RBD beads for 1 h. Beads were collected by centrifugation, washed and resuspended in loading buffer. All samples were boiled for 5 min and resolved by SDS-polyacrylamide gel electrophoresis. RhoA was detected by immunoblot.

**Confocal fluorescence microscopy and image quantification.** Cells were seeded on top of a collagen I matrix and immunostained as described[12]. Cells were fixed with 4% formaldehyde, permeabilised with 0.3% Triton X-100, blocked with 5% bovine serum albumin (BSA), and stained with primary antibody pSer19-MLC2 (1:200, #3671, Cell Signalling), which was detected with secondary Alexa Fluor 488

or 647 antibodies (Life Technologies). F-actin was stained using Alexa Fluor 546-phalloidin (Life Technologies) and nuclei with Hoechst 33258 (Life Technologies). Antibodies were diluted in 5% BSA-PBS. Imaging was carried out on Zeiss LSM 510 Meta confocal microscope with Plan-Apochromat 40x/1.2 NA (water) objective lenses and Zeiss LSM 710 confocal microscope with Plan-Apochromat 40x/1.3 Oil or a Plan-Apochromat 63x/1.4 NA (oil) objective lenses (Carl Zeiss) and Zen software. Images were analysed using ImageJ. p-MLC2 fluorescence signal was quantified calculating the pixel intensity in single cells relative to the cell area.

**Adhesion assays to keratinocyte monolayer.** HaCaT keratinocytes were seeded in a 96-well plate at $10^5$ cells/well and, 24 h later, melanoma cells stained with CMFDA green dye (10 μM, Life Technologies) were seeded on top at $10^4$ cells/well. After 2 h at 37 °C, cells were imaged. Plates were then washed twice with PBS and re-imaged. Percentage of adhering cells was calculated as the number of cells imaged after vs before washing.

**3D invasion assays.** Cells were suspended in serum-free bovine collagen I at 2.3 mg/ml to a final concentration of $1 \times 10^4$ cells per 100 μl of matrix, seeded on 96-well plates and spun down to the bottom of the well. After matrix polymerization, 10% FBS-containing media was added on top of the matrix, allowing the cells to invade upwards for 24 h as previously described[8,14]. Then, plates were fixed in 4% formaldehyde, stained with 5 μg/ml Hoechst 33258 (Life Technologies) and imaged on Zeiss LSM 510 or Zeiss LSM 710 confocal microscopes (Carl Zeiss) with Zen software. Confocal z-slices were collected from each well at the bottom of the well and at 50 μm. The 3D invasion index was calculated as number of invading cells at 50 μm divided by the total number of cells.

**Intracellular measurement of ROS.** Cells were harvested and incubated with 5 μM CellROX Green Reagent (Life Technologies) at 37 °C for 30 min in the dark. Then, cells were diluted with FACS buffer (PBS−, 1% BSA, 2 mM EDTA, 0.1% NaN₃) and fluorescence was analysed by flow cytometry using a BD LSRFortessa™ system (Flow Cytometry Core, Barts Cancer Institute) and FlowJo software (Supplementary Fig. 7k). Cells were first gated on FSC-A vs SSC-A to eliminate cell debris. Then, discrimination of doublets was performed on FSC-A vs FSC-H and SSC-A vs SSC-W. Only viable and single cells were used for the analysis. CellROX green positivity was analysed using mean fluorescence intensity.

**Cell viability assay.** Cells stably expressing shRNAs or cells after 48 h transfection with siRNAs were seeded in 6-well plates (25,000 cells/well). For ROCKi experiments, 25,000 cells were seeded in 6-well plates, allowed to adhere and treatment was performed as indicated for each experiment. Cell viability was assessed after 7 days in all experiments, while for *MYL9* and *MYL12B* siRNA experiments that it was assessed after 3 and 7 days. Then, cells were fixed with 1% formaldehyde and stained with 0.25% crystal violet. After air-drying, crystal violet stain was dissolved in 10% acetic acid. Samples from each well were transferred to a 96-well plate and the absorbance was measured at 595 nm on a microplate reader as an indirect measure of cell number. Results were normalized to the value of initially plated cells and presented as percentage of cells in each sample relative to control.

**Quantitative real time one-step PCR and qPCR EMT array.** Total RNA was isolated using TriZol (Life technologies) and RNeasy Mini Kit (Qiagen). qPCR primers and Brilliant III Ultra-Fast SYBR Green QRT-PCR Master Mix (Agilent Technologies) with 50 ng RNA were used following the manufacturer's instructions.

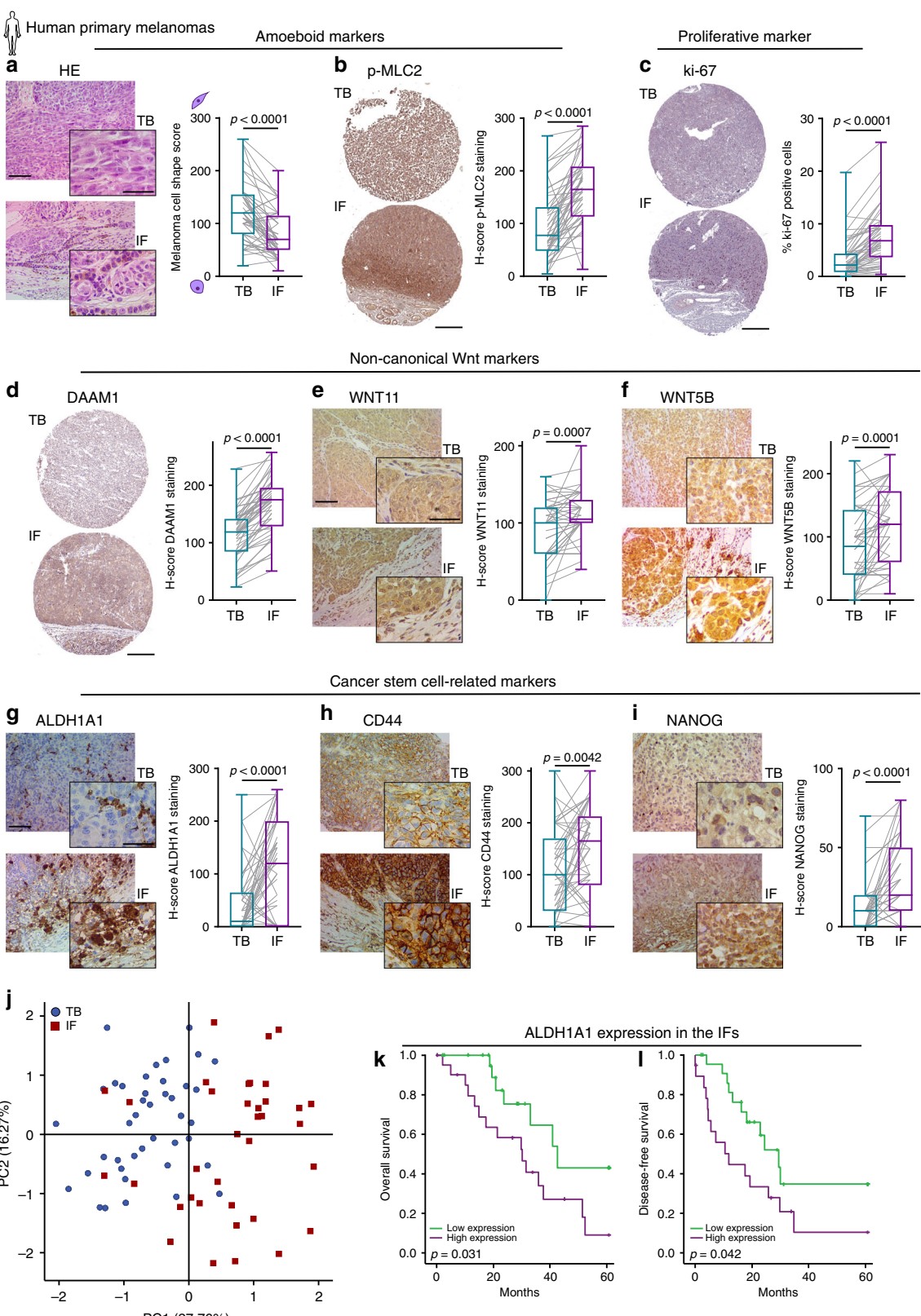

*GAPDH* was used as loading control. The following qPCR primers from Sino Biological were used: *WNT11* (HP101586), *WNT5B* (HP101843), *ALDH1A1* (HP100088), *OCT4* (HP101756). The following QuantiTect Primers from Qiagen were used: *GAPDH* (QT00079247), *ROCK1* (QT00034972), *ROCK2* (QT00011165), *MYL9* (QT00072268), *MYL12B* (QT00075264), *FZD7* (QT00213850), *RYK* (QT00047292), *PLCB1* (QT00038689), *DAAM1* (QT00005579), *ALDH1A3* (QT00077588), *NANOG* (QT01025850), *JARID1B* (QT00060648), *CD44* (QT00073549), *SOX2* (QT00237601), *ALCAM* (QT00026824).

Epithelial to mesenchymal transition (EMT) RT2 Profiler PCR Array (330231, PAHS-090Z, Qiagen) was performed according to manufacturer's instructions. For that, 2 µg of total RNA were reverse-transcribed to cDNA using RT2 First Strand Kit (Qiagen) and subjected to qPCR EMT array. Reactions were run on a QuantStudio 7 Flex Real-Time PCR System (ThermoFisher Scientific). PCR array was analysed using the web-based tool (https://dataanalysis.qiagen.com/pcr/arrayanalysis.php) supplied by the manufacturer and gene expression was normalized to housekeeping genes.

**Fig. 7 Analysis of the invasive front of human primary melanomas. a–i** Representative images (left) and quantification (right) of **a** melanoma cell shape score, **b** H-score of p-MLC2 staining, **c** ki-67 positive cells, H-score of **d** DAAM1, **e** WNT11, **f** WNT5B, **g** ALDH1A1, **h** CD44 and **i** NANOG staining in matched TB and IF from primary melanomas. **j** Principal component analysis (PCA) based on the expression of amoeboid (cell shape and p-MLC2), proliferative (ki-67), non-canonical Wnt pathway (WNT11, WNT5B and DAAM1) and cancer stem cell-related (ALDH1A1, CD44 and NANOG) markers assessed by immunohistochemistry in matched TB and IF from primary melanomas. Percentage of variation explained by each component is given in the axis labels. **k, l** Kaplan–Meier survival curves of **k** overall survival and **l** disease-free survival according to ALDH1A1 protein expression in the IF from our cohort of primary melanomas. ALDH1A1 expression was categorized as low or high using the median expression. **a–l** 53 primary melanomas. **a, e–i** Scale bar, 100 μm; inset, 50 μm. **b–d** Scale bar, 300 μm. **a–i** Box limits show 25th and 75th percentiles, the horizontal line shows the median, and whiskers show the minimum and maximum range of values. **a, b, d–f** Two-tailed paired t-test. **c, g–i** Two-tailed Wilcoxon test. **j** Scatter plot showing principal components 1 (PC1) and 2 (PC2). **k, l** Log-rank test. Human schematic in this figure was created using Servier Medical Art templates licensed under a Creative Commons Attribution 3.0 Unported License (https://smart.servier.com).

**Analysis of gene expression data from human databases.** From the public database GEO (Gene Expression Omnibus), Xu GSE8401[35] (31 primary and 52 metastatic melanomas), Riker GSE7553[34] (14 primary and 40 metastatic melanomas) and Kabbarah GSE46517[33] (31 primary and 73 metastatic melanomas) series were extracted. Data were normalized using GenePattern platform (https://www.broadinstitute.org/cancer/software/genepattern).

Gene expression data and clinical information of human melanoma samples (70 primary and 319 metastatic melanomas) from The Cancer Genome Atlas (TCGA) database were downloaded from Firehose (https://gdac.broadinstitute.org/). Only TCGA samples with no neo-adjuvant treatment prior to tumour resection were considered.

**Gene enrichment analyses.** Normalized gene expression microarray data of amoeboid melanoma cells (GSE23764)[10] was analysed by comparing amoeboid A375M2 cells to more elongated and less contractile A375P cells or to A375M2 cells treated with ROCK1/2 inhibitors (H1152 and Y27632) or blebbistatin. Gene sets for proliferative and invasive signatures from Verfaille[19] and Hoek[18] studies; stem cell-like signatures and EMT processes were downloaded and analysed using Gene Set Enrichment Analysis (GSEA) software[68] (http://www.broadinstitute.org/gsea/index.jsp) with the specific settings: permutations-1,000, permutation type-gene set, metric for ranking genes-t-test. For single-sample Gene Set Enrichment Analysis (ssGSEA), significantly enriched stem cell-like/EMT gene sets in amoeboid A375M2 cells were considered according to p-value < 0.05 and FDR < 0.25 in at least 3 of the 4 comparisons performed. To calculate the gene-signature score in each sample, we used ssGSEA Projection Software[68,69] from GenePattern platform (https://www.broadinstitute.org/cancer/software/genepattern). Heatmaps were generated using MeV_4_9_0 software (http://mev.tm4.org/).

Network enrichment analysis of significantly downregulated genes in at least one cell line also downregulated or with no expression changes in the other cell line from qPCR EMT array was performed using Ingenuity Pathway Analysis (Qiagen).

**Animal studies.** All animals were maintained under specific pathogen-free conditions and handled in accordance with the Institutional Committees on Animal Welfare of the UK Home Office (The Home Office Animals Scientific Procedures Act, 1986). All animal experiments were approved by the Ethical Review Process Committees at Barts Cancer Institute, King's College London and The Francis Crick Institute and carried out under licences from the Home Office, UK. All mice were obtained from Charles River UK. Mice used were 6–12 weeks old. Animals were housed in groups of 4–5 mice per cage with access to food and water ad libitum. Mice were maintained on an alternating 12 h light-dark cycle, with controlled room temperature (21 ± 1 °C) and relative humidity (40–60 %).

For tumour initiation assays with pre-treated A375M2 and B16F10 cells, prior to injection cells were pre-treated in vitro with ROCK1/2 inhibitors (5 μM H1152, 1 μM or 5 μM GSK269962A) for 5 days, with fresh drug added every day. Then $1 \times 10^6$ A375M2 cells or $2 \times 10^5$ B16F10 cells were mixed in 100 μl of Growth Factor Reduced Matrigel (356230, Corning), and injected subcutaneously into female NOD/SCID/IL2Rγ− (NSG) mice (A375M2) or C57BL/6 J mice (B16F10). Mice were not treated during the course of the experiment.

For limiting dilution assays, $5 \times 10^5$, $5 \times 10^4$, $1 \times 10^4$ or $5 \times 10^3$ cells (A375M2/A375P cells, stable shFZD7#1/shDAAM1#1/shControl WM1361 cells and stable shDAAM1#1/shControl A375M2 cells) were mixed in 100 μl of Growth Factor Reduced Matrigel (356230, Corning) and injected subcutaneously into female NSG mice. Tumour growth was assessed 3 weeks after injection. The tumour-initiating frequency (TIF) was estimated by ELDA[22] using the bottom 20th percentile of tumour size in each experiment as the limiting threshold for positive growth. Only tumours above this threshold were considered.

For the orthotopic experiment using 4599 cells, $2 \times 10^5$ 4599 cells stably expressing hNIS-mCherry (4599.hNIS-mCherry) in 30 μl PBS were injected intradermally into male NSG mice. Tumours were allowed to establish and treatment started at day 11 when mean tumour volume was 20 mm³. Drug administration (25 mg/kg GSK269962A dissolved in 5% DMSO or vehicle 5% DMSO) was performed daily by oral gavage.

For all the above tumour models, tumour size was monitored and determined by caliper measurements and calculated as tumour volume (mm³) = length × width × height × 0.52. At the end of experiments, prior to dissection tumours were measured; or dissected tumours were weighed. Tumours were formalin-fixed and paraffin-embedded (FFPE) for immunohistochemical analysis.

For experimental metastasis assays, $1 \times 10^6$ stable shDAAM1 and shControl WM1361 cells in 100 μl PBS were injected into the tail vein of NSG mice (matched male and female). Mice were sacrificed after 30 min (to show that equal cell numbers reach the lung), 24 h and 3 weeks, and then lungs were washed, fixed in 4% formaldehyde for 16 h and examined for fluorescent signal (GFP from shRNA constructs) under a Zeiss LSM 510 Meta confocal microscope (Carl Zeiss). 20–25 images per mouse lung were analysed, and lung retention was represented as percentage of fluorescence area per mouse lung.

For experimental metastasis assays using 4599 cells, $4 \times 10^5$ cells pre-treated in vitro with 5 μM GSK269962A for 24 h were injected in 200 μl PBS into the tail vein of NSG female mice. Mice were treated daily (25 mg/kg GSK269962A dissolved in 5% DMSO or vehicle 5% DMSO) by oral gavage. Mice were sacrificed after 30 min and 6 days, and then lungs were washed, fixed and examined by immunohistochemistry. Similarly, $4 \times 10^5$ 4599 cells without any pre-treatment in 200 μl PBS were injected into the tail vein of NSG female mice and animals were treated daily (25 mg/kg GSK269962A dissolved in 5% DMSO or vehicle 5% DMSO) by oral gavage for 12 days.

**Immunohistochemistry**

*Case selection.* Two tissue microarrays, including FFPE biopsies of 53 human primary melanomas and 45 metastases (Supplementary Tables 2 and 3) were included in the case series. Each biopsy was represented by two cores (1 mm diameter) from the tumour body (TB) and two cores from the invasive front (IF) areas. The IF was defined as melanoma cells with at least 50% contact with the matrix as previously described[7,10,12]. Tumours were classified following the most recent World Health Organization criteria. Tumour samples were processed by IRBLleida (PT17/0015/0027) and HUB-ICO-IDIBELL (PT17/0015/0024) Biobanks integrated in the Spanish National Biobank Network and Xarxa de Bancs de Tumors de Catalunya following standard operating procedures with the appropriate approvals from the Ethics and Scientific Committee. Samples were collected with specific informed consent, in accordance with the Helsinki Declaration.

Whole sections from subcutaneous tumours (A375M2/A375P tumours, shControl/shFZD7/shDAAM1 WM1361 tumours and shControl/shDAAM1 A375M2 tumours from limiting dilution assays), from intradermal tumours (4599 cells) and mice lungs from tail vein metastasis assays (4599 cells) were included.

Melanospheres from serial sphere passages were also included. Agar pellets were generated with spheres from every passage. Spheres were collected gently and spun down at $300 \times g$ for 10 min. Pellets were washed twice with PBS and embedded in 50 μl 3% agar. Finally, agar pellets were embedded in paraffin using standards methods for immunohistochemical analysis.

*Experimental procedure.* All tissue samples were FFPE, sectioned (3–4 μm-thick) and dried for 1 h at 65 °C. Next, tissue samples were subjected to deparaffinization, rehydration and heat-induced epitope retrieval using a Biocare Decloaking Chamber (DC2012) at 110 °C for 6 min in the corresponding unmasking solution (Supplementary Table 5). The endogenous peroxidase was blocked with dual endogenous enzyme-blocking reagent (#S2003, Agilent) for 10 min. Reagents were incubated at room temperature in a humidified slide chamber. Detailed information about the antibodies, dilutions and conditions used for the immunohistochemical stainings are specified in Supplementary Table 5. All the stainings were counterstained with Haematoxylin.

*Imaging and scoring.* Morphologic analysis of tumour tissues was performed on haematoxylin- and eosin-stained (H&E) sections as previously described[10]. Cell shape was graded from 0 to 3 (0 = round, 1 = ovoid, 2 = elongated and 3 = spindly) and a score was assigned as follows: cell shape score = ((percentage of cells [%] shape 0 × 0) + (% shape 1 × 1) + (% shape 2 × 2) + (% shape 3 × 3)), with values ranging from 0 (all cells round) to 300 (all cells spindle). Using consecutive

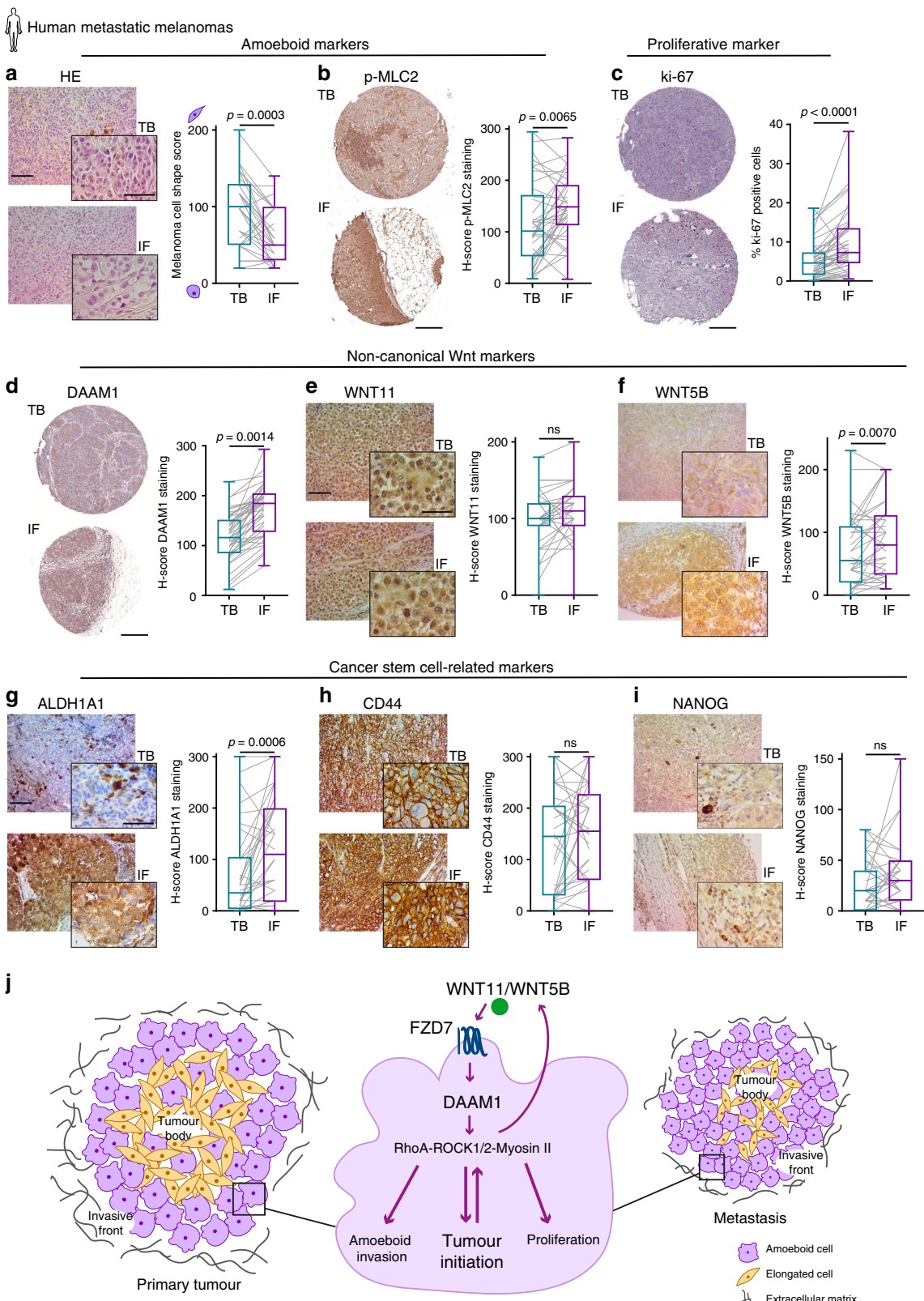

sections, IHC staining for all the markers was graded semiquantitatively by considering the percentage and intensity of the staining. A histologic score (H-score) was obtained from each sample, and values ranged from 0 (no immunoreaction) to 300 (maximum immunoreactivity). The score was obtained by applying the following formula, H-score = 1 × (% light staining) + 2 × (% moderate staining) + 3 × (% strong staining). WNT11, WNT5B, CD44, NANOG, OCT4, ALDH1A1 and ALDH1A3 stainings were scored blind and a staining H-score was provided for each section. Representative pictures according to mean H-score were imaged

using iScope microscope (IS.1159EPLi, Euromex) and ImageFocus 4.0 (Euromex) software. For p-MLC2, ki-67, FZD7 and DAAM1 stainings, whole section images (WSI) were scanned using NanoZoomer S210 slide scanner (Hamamatsu). Staining quantification was performed with QuPath 0.1.2 software[70]. Positive cell detection was performed, and three different thresholds were applied according to the intensity scores (0, 1, 2 and 3). Next, software was trained creating random tree classification algorithms combined with intensity information in order to differentiate tumour cells from necrosis, stroma and immune cells. Then, IHC staining

**Fig. 8 Analysis of the invasive front of human metastatic melanomas. a–i** Representative images (left) and quantification (right) of **a** melanoma cell shape score, **b** H-score of p-MLC2 staining, **c** ki-67 positive cells, H-score of **d** DAAM1, **e** WNT11, **f** WNT5B, **g** ALDH1A1, **h** CD44 and **i** NANOG staining in matched TB and IF from melanoma metastases. **j** Model summarizing the findings of this study. The IF of human primary melanomas is enriched in cells with rounded-amoeboid morphology, high levels of Myosin II, proliferative, non-canonical Wnt and cancer stem cell-related markers. This phenotype is recapitulated and further enriched in melanoma metastasis. WNT11/5B activate FZD7 and DAAM1 to control Rho activity and then ROCK1/2-Myosin II levels. All these signalling components have an impact on amoeboid features and tumour initiation in melanoma both in vitro and in vivo. Specifically, DAAM1 plays an essential role in tumour and metastasis initiation and subsequent metastatic outgrowth by sustaining the amoeboid phenotype. **a–i** 45 metastatic melanomas. **a**, **e–i** Scale bar, 100 μm; inset, 50 μm. **b–d** Scale bar, 300 μm. **a–i** Box limits show 25th and 75th percentiles, the horizontal line shows the median, and whiskers show minimum and maximum range of values. **a**, **b**, **d–f** Two-tailed paired *t*-test. **c**, **g–i** Two-tailed Wilcoxon test. Human schematic in this figure was created using Servier Medical Art templates licensed under a Creative Commons Attribution 3.0 Unported License (https://smart.servier.com).

was graded semiquantitatively by considering the percentage and intensity of the staining, and H-score values were calculated as described above. Furthermore, to show spatial correlation between different markers (ALDH1A1, CD44, p-MLC2, DAAM1 and ki-67), hotspots maps were created using the optical density mean (OD mean)—Chanel 2: VIP stain intensity in QuPath. For co-localization analysis, images for p-MLC2 and ki-67 markers were aligned in FIJI v1.52p using TrackEM2 module. Next, colour deconvolution was performed using AEC-Haematoxylin vectors and a composite was created using channel-2 (red) for each staining. The composite was adjusted inverting the LUT for each marker and was given a pseudocolour. mCherry staining was performed for quantification of invading cells in primary tumours from experiments using 4599 cells. WSI were scanned using NanoZoomer S210 slide scanner (Hamamatsu). mCherry analysis was performed using QuPath software: an invasive area was delimited underneath the tumour invasive front and positive cells were counted using the positive cell detection plug-in. Assessment of lung colonization assays with 4599 cells was performed on H&E WSI of lungs using QuPath software. Several supervised annotations were performed in each study identifying tumour, normal tissue and whitespace areas. Next, pixel classifier was performed in the whole-image for all the cases using random-tree classification and low resolution.

**Statistical analysis**. Unpaired *t* test, Mann–Whitney's test, paired *t*-test, Wilcoxon test, one-way ANOVA with Dunnett or Tukey post hoc test or Benjamini, Krieger and Yekutieli correction, and Kruskal–Wallis with Dunn's multiple comparison test were performed using GraphPad Prism (GraphPad Software, Inc). Principal Component Analysis, survival curves estimation based on the Kaplan–Meier method and log-rank test were performed using SPSS Statistics (IBM). Limiting dilution assays were analysed using ELDA software (http://bioinf.wehi.edu.au/software/elda/)[22]. Data were plotted as graphs showing mean ± standard error of the mean (s.e.m.); or boxplots where box limits show 25th and 75th percentiles, the horizontal line shows the median, and whiskers show minimum and maximum range of values as indicated in figure legends. *P* values were calculated using two-tailed tests. *P* values of less than 0.05 were considered statistically significant. In Figure legends, "*n*" means number of independent biological experiments unless otherwise stated.

**Reporting summary**. Further information on research design is available in the Nature Research Reporting Summary linked to this article.

## Data availability

Gene expression datasets re-analysed in this study are available from NCBI GEO under accession numbers: GSE23764, GSE8401, GSE7553 and GSE46517. Uncropped immunoblot scans for representative images displayed in the figures are shown in Source data. All other relevant data supporting the findings of this study are available within the article and its Supplementary information and from the corresponding author on reasonable request. Source data are provided with this paper.

## Code availability

No custom codes were used in this study. All codes are indicated in the appropriate Methods sections and references.

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

## Acknowledgements

This work was supported by Cancer Research UK (CRUK) C33043/A12065 and C33043/A24478 (V.S.-M., I.R.-H., O.M., L.K., G.C. B.F. and J.L.O.); Royal Society RG110591 (V.S.-M.); Barts Charity (V.S.-M., I.R.-H., O.M., J.M. and J.L.O.); Fundacion Alfonso Martin Escudero and Marie Sklodowska-Curie Action, grant agreement No 659022 (I.R.-H.); MRC C97993H (G.C.); The Harry J. Lloyd Charitable Trust (J.L.O. and V.S.-M.); Francis Crick Institute core funding from CRUK FC001112, MRC FC001112 and the Wellcome Trust FC001112 (I.M. and A.P.); CRUK C48390/A21153, CRUK/EPSRC and Wellcome Trust/EPSRC WT 203148/Z/16/Z (G.O.F.); NIHR BRC at Guy's and St Thomas' NHS Foundation Trust and KCL IS-BRC-1215–20006, CRUK C30122/A11527 and C30122/A15774, MRC MR/L023091/1, CRUK/NIHR in England/DoH for Scotland, Wales and Northern Ireland ECMC C10355/A15587 (S.N.K.); ISCIII/FEDER "Una manera de hacer Europa" FIS-PI1500711 and PI18/00573 (R.M.M.); CIBERONC CB16/12/0023 (R.M.M and X.M.-G). Views expressed are those of the author(s) and not necessarily those of the NHS, the NIHR, or the Department of Health.

## Author contributions

V.S.-M. was principal investigator, designed the study, supervised experiments, and wrote the paper; I.R.-H. designed the study, performed most experiments, and wrote the paper. O.M., L.K., G.C., A.P., J.M., B.F., V.L.B. and J.L.O. performed experiments; R.M.P., J.M., R.M.M. and X.M.-G. provided human tissue samples; S.N.K. supervised the in vivo experiment performed by J.L.O.; G.O.F. supervised in vivo experiments performed by B.F.; I.M. designed the study and supervised in vivo experiments performed by A.P. and V.L.B.

## Competing interests

The authors declare no competing interests.
