## [Peer Review File · Nature Communications]

Reviewers' comments:

Reviewer #1 (Remarks to the Author); expert in Wnt signalling,

This is a potentially interesting paper about non-canonical Wnt signalling controlling the so-called "amoeboid state" of melanoma cells. The authors propose that this state is associated with stemness features and the capacity of melanoma cells to both invade/metastasize and proliferate. WNT11, its receptor FZD7, and the intracellular signalling molecule DAAM1 are reported to control the behaviour of melanoma cells in vitro and in vivo by regulating amoeboid behaviour.

This is a follow-up story of a recent publication by the same group on amoeboid melanoma cells and their interaction with the immune environment (Cell 2019, 176:757-774). Here, the authors describe mechanisms controlling cell-intrinsic features contributing to the progression of the disease. The data are novel and relevant and might deserve publication in this journal, should the authors be able to address the issues specified below.

1) At several places, the authors refer to 'stemness features' of amoeboid cells. What exactly do they mean by this? The only 'stemness' assay they use is sphere formation and self-renewal in vitro. Others have shown self-renewal of particular states/cell subpopulations in vivo. Moreover, some groups have used 'tumor cell stemness' and 'tumor initiation capacity' as equivalent terms. In the latter cases, the tumor initiation capacity was assessed at clonal dilution, which the authors don't do here. Importantly, since the authors don't isolate specifically the amoeboid state (but rather use cell lines that harbour, among others, cells with the capacity to acquire amoeboid features), 'stemness' of amoeboid cells has not been shown by the authors.

2) Based on gene expression studies at the population level and on WNT11 kd assays (which affect invasiveness without perturbing proliferation), they conclude that amoeboid cells are invasive and at the same time proliferative, with "no need for phenotype switching". Again, since the authors don't resolve the amoeboid state at the single cell level, the claim that cells in an amoeboid state are both invasive and proliferative cannot be made.

3) Likewise, the authors propose that ROCK-Myosin II and WNT-FZD-DAAM1 control stemness and invasiveness in amoeboid cells. What's the evidence that these pathways specifically affect cells in an amoeboid state?

4) For many of their experiments, the authors use only one siRNA (or shRNA) to address the function of a given gene. They have to repeat these experiments with at least two siRNAs/shRNAs per gene.

5) Assessing the tumorigenic capacity of shDAAM1 melanoma cells in vivo is one of the key experiments of this study (Fig. 5). This experiment has to be repeated with additional cell lines.

6) Related to my point 2), proliferation has to be assessed in the experiments using ROCKi (Fig. 2, in vitro and in vivo) and shDAAM1 (Fig. 5, in vitro and in vivo). Similarly, proliferation has to be assessed in the in vitro experiments presented in Figs. 1b, 1d, and 3h.

7) Non-amoeboid cell lines display increased self-renewal capacity over time, which is associated with acquisition of amoeboid features (Fig. 1). Accordingly, cell lines with amoeboid features (eg A375M2 and others) should have an intrinsically higher self-renewal and tumor-initiation capacity than non-amoeboid cell lines. Is this the case?

8) The role of FZD7 has to be addressed in vivo in order to validate the relevance of the WNT11/FZD7/DAAM1 signalling axis (Fig. 4).

9) The link from the experiments addressing the role of ROCK and WNT11/FZD7/DAAM1 (Figs 1 to

5) to the rather descriptive experiments presented in Figs. 6, 7 is unclear. Importantly, whether cells with high levels of DAAM1, WNT11 also express high ALDH1A1, NANOG, pMLC2, etc, has to be shown at the cellular level. Otherwise, it cannot be stated that "amoeboid cells are linked to poor clinical outcome" (a statement solely based on ALDH1A1 TCGA data; Fig. 6k).

10) In several figures, the data are presented as bar graphs without showing the actual data points. These should be included, particularly because for many experiments the error bars and data point distribution seem to be huge. In Fig. 1g, for instance, most data points for relative p-MLC2 levels appear to be very similar in GFP vs MLC2, suggesting that only a minor fraction of data points accounts for the significance in differential p-MLC2 expression. Do the authors have an explanation for the apparently differential responsiveness to MLC2-GFP expression in melanoma cells?

11) In Fig. 2, p-MLC2 levels have to be assessed at the end point of the experiments (i.e. 18 days for the A375M2 and 11 days for the B16F10 line).

12) On p.7, the authors claim that "ROCKi leads to loss of cell rounding and decreased p-MLC2 levels in all areas" of the tumor. However, this doesn't seem to be the case for the TB area (Fig. 2g, h).

13) How exactly do the authors determine the melanoma cell shape score and the roundness index? This should be better explained.

14) In Fig. 2o, decreased metastasis of ROCKi-treated tumors is interpreted as decreased invasiveness/metastatic capacity. However, decreased numbers of metastases at distant sites might simply be due to the decreased size of the primary tumors.

15) Do the authors have any explanation for how ROCK might control EMT-related genes, including WNT11? This should at least be discussed.

Reviewer #2 (Remarks to the Author); expert in melanoma and stem cells:

Melanoma stemness is fueled by Wnt-DAAM1 signaling in the amoeboid invasive compartment

The authors herein report interesting findings revealing the role of amoeboid melanoma cells in stemness and metastasis. The authors identify a Wnt-DAAM1 axis as critical for the aggressive phenotypes of amoeboid cells, providing rationale to therapeutically target this population of melanoma cells. The findings will be of particular interest to the melanoma field, however the below issues will need to be addressed before publication:

- The abstract needs rewriting for clarity
- Figure 1B: the authors use A375P and WM983A because they have been previously reported to have low levels of Myosin II.
 - o The authors should here repeat WB or RT-PCR of Myosin II in these cell lines to demonstrate how robust this is.
 - o Two cell lines that have high Myosin II levels should also be used to demonstrate the importance of Myosin II levels in the discussed phenotypes
- The authors should probe for amoeboid markers by WB or RT-PCR following the serial passaging in low adherent conditions (Figure 1B) and collagen I (Figure 1C) to strengthen their scientific point.
- For Figure 1G, show the WB for the phospho-MLC2 levels for the reader to appreciate

- Why do the authors use an NRAS-MT cell line (WM1361) in Figure 1K. What is the justification for which genotype of melanoma to focus on?
- What are the impacts on melanoma viability from using ROCK inhibitors and silencing ROCK and MLC2 genes?
- How were A375M2 cells generated? The authors unfortunately use only 1 amoeboid cell line for many experiments and it is difficult to appreciate how robust these findings would be in an expanded cell line panel.
- Figure 2, A365M2 cells were treated for 5 days before in vivo transplantation with ROCKi. What is the impact on melanoma cell viability after this treatment?
 - o If ROCKi causes extensive cytotoxicity in vitro, obviously an impact on in vivo growth would be observed
- Figure 2O,P: It is difficult to determine whether the ROCK inhibitor decreases spontaneous metastases or the ROCKi pretreatment before injection into mice kills all melanoma cells outright.
- Figure 3C needs statistics. Which of these changes in gene expression are significant?
- Figure 3E: It is confusing why there are also stars indicating statistical significance achieved in cells treated with no (-) siRNA?
 - o There does not appear to be a significant difference between any condition relative to the (-) control. The same goes for Figure 3F. The authors should show every replicate as a dot.
- Figure 4 C, Show every measurement as an individual dot. It is difficult to see how the data shows statistical differences by the way the data is currently displayed.
- Are the data shown in 4D and 4E showing only statistically significant changes? Need statistics.

Reviewer #3 (Remarks to the Author); expert in cell migration and morphodynamics:

Rodriguez-Hernandez and coauthors present extensive evidence from in vitro experiments, several in vivo models, patient tissue samples, and previously published datasets to describe a novel pathway in melanoma whereby non-canonical soluble Wnts (Wnt11 and Wnt5b) induce myosin II activity through an FZD7-DAAM1-ROCK signaling axis. In cell culture experiments with melanoma cell lines, heightened myosin activity as evidenced by phosphorylated MLC (pMLC) was associated with a rounded cell morphology reminiscent of cells migrating in an amoeboid fashion, higher capacity to form melanospheres, and expression of genes associated with stemness and EMT. Melanoma cells transplanted into mice formed tumors with rounded cells at the tumor margin and showed potential for metastasis to the lung, but pre-treatment with a ROCK inhibitor diminished these features. The authors used siRNA/shRNA to flesh out the pathway showing that autocrine or paracrine Wnt ligands engage the FZD7 receptor, signal through Dvl-DAAM1-Rho, and induce downstream myosin activity associated with a more rounded cell morphology. Having established these key components and after implicating this pathway in tumor growth and metastasis in vivo, they then histologically examine human primary and metastatic melanoma tissue for markers of stemness, non-canonical Wnt markers, and amoeboid markers (pMLC and shape) showing that in both locations these markers are heightened at the tumor margin or invasive front. In sum, the authors put forth that an invasive front enriched with amoeboid cells with expression profiles indicative of both stemness and EMT are a key reason for the aggressiveness of malignant melanomas and a potential prognostic indicator. This body of work implying a simultaneous migratory and stem-like cancer cell phenotype is deemed novel given current notions of cancer stem cell function and location within the tumor and previous work from Friedl and colleagues on the topic of amoeboid-mesenchymal transition. The work provides new insights and is extensive, and merits publication in Nature Communications, but a few concerns are listed below:

Major:

1. A major implication of the work is that cells with an amoeboid phenotype are linked to the aggressiveness of melanoma. The term amoeboid is a descriptor of a cell's migratory mode and not just shape – conventionally involving proteolytic-independent migration with limited adhesion to the ECM, high cytoskeletal contractility, cytoskeletal extensions via short protrusions or blebs,

and a rounded (unspread) morphology. The authors heavily rely on this last descriptor (rounded cell morphology) to implicate amoeboid migration in their findings, but do not directly assess migration phenotype. Assays for attachment to keratinocytes and invasion into collagen matrices were performed (Figure 3), but neither of these directly supports a central role for amoeboid migration in the presented studies. While this is obviously quite difficult to obtain from in vivo studies, in vitro studies (either timelapse or fixed time-point as in Figure 3h demonstrating invasion in the presence of an MMP inhibitor) would strengthen this link.

2. In emphasizing the involvement of amoeboid migration, the authors perform experiments to show that the proliferative capacity of these cells is unaffected by the Wnt-FZD7-DAAM1-ROCK-MLC pathway. They seem to imply that the amoeboid migratory phenotype is a driver of tumor growth and spread, de-emphasizing the role of proliferation which presumably is critical to melanosphere formation and primary tumor growth. Given many previous connections between myosin activity, cytoskeletal contractility, and proliferation, one would expect the ROCK inhibited cells in the studies presented in Figure 2 to proliferate less. Related to this, the quantification of proliferation isn't clear: as described in the Methods, crystal violet was used to count cells after 3-7 days and this value was normalized to the value of initially plated cells. All of the provided quantifications are at or near 100% - does this imply cells did not proliferate over this period (which should be specified more precisely)? The authors should clarify in the text the role of proliferation (perhaps de-emphasizing amoeboid migration given comment #1 above). Additionally, including a better assessment of proliferation such as EdU incorporation or Ki67 staining performed in their sphere formation assays (eg. in comparing A375P vs A375M2, or with contractility perturbations) would provide a clearer sense of how proliferation is altered and potentially contributing to in vivo outcomes.

3. Can the authors comment on how they can rule out the influence of ROCK inhibition on initial engraftment in in vivo studies? Given the role of myosin/contractility in adhesion, would could imagine that ROCK inhibition leads to poor initial attachment and thus significant cell death immediately following injection.

Minor:

1. The title could confuse many readers as "amoeboid invasive compartment" is not an established term and what the compartment is or where it's location is are both ambiguous. Additionally, given comment #1 above, if the mode of cell migration is not shown to amoeboid, a title change would be warranted.

2. The quantification for cell shape in vitro is not sufficiently described, despite the provided range of 0 to 1. Including an equation would be helpful. Related to this, showing cell outlines (of ~10-20 cells) throughout would be more convincing than fluorescent images of only 1-3 cells.

3. Labeling of controls as "-" in Figure 3b westerns is slightly confusing.

4. Figure 3e "****" above control - presumably the indicated significances above other groups are comparisons to the control?

5. Do the pMLC2 results change when local variations in cell density are accounted for? An unspread, rounded cell morphology would generally lead to higher density of any stained cytosolic protein compared to spread cells that are additionally more separated from each other.

6. The experiment presented in Figure 2m-p is not sufficiently described in the text or figure legend.

Reviewers' Comments:

We thank all Reviewers for their comments, we believe our work has greatly improved after addressing all of them.

Rev#1 Expert in Wnt signalling

This is a potentially interesting paper about non-canonical Wnt signalling controlling the so-called “amoeboid state” of melanoma cells. The authors propose that this state is associated with stemness features and the capacity of melanoma cells to both invade/metastasize and proliferate. WNT11, its receptor FZD7, and the intracellular signalling molecule DAAM1 are reported to control the behaviour of melanoma cells *in vitro* and *in vivo* by regulating amoeboid behaviour.

This is a follow-up story of a recent publication by the same group on amoeboid melanoma cells and their interaction with the immune environment (Cell 2019, 176:757-774). Here, the authors describe mechanisms controlling cell-intrinsic features contributing to the progression of the disease. The data are novel and relevant and might deserve publication in this journal, should the authors be able to address the issues specified below.

1) At several places, the authors refer to ‘stemness features’ of amoeboid cells. What exactly do they mean by this? The only ‘stemness’ assay they use is sphere formation and self-renewal *in vitro*. Others have shown self-renewal of particular states/cell subpopulations *in vivo*. Moreover, some groups have used ‘tumor cell stemness’ and ‘tumor initiation capacity’ as equivalent terms. In the latter cases, the tumor initiation capacity was assessed at clonal dilution, which the authors don’t do here. Importantly, since the authors don’t isolate specifically the amoeboid state (but rather use cell lines that harbour, among others, cells with the capacity to acquire amoeboid features), ‘stemness’ of amoeboid cells has not been shown by the authors.

We apologize for the terminology used in the previous version of the manuscript. Originally, we have used sphere formation and self-renewal assays and stem cell-related gene expression analysis in order to investigate the *in vitro* tumour-initiating potential of amoeboid melanoma cells.

We have expanded these observations using limiting dilution assays where we have now also assessed the *in vivo* tumour initiation capacity of A375M2 cells (90-95% rounded with high Myosin II levels) compared with more elongated A375P cells (50% rounded, 50% elongated with lower Myosin II levels) (Sanz-Moreno et al, Cancer Cell 2011; Orgaz et al, Nat Commun 2014; Georgouli et al, Cell 2019) (new Fig.1b-e and Supplementary Fig.1a,c). Amoeboid A375M2 cells were specifically derived and isolated from A375P cells after two rounds of lung metastatic colonization (Clark et al., Nature 2000). We show now that- *in vivo*-amoeboid A375M2 cells exhibit 5-fold higher tumour-initiating frequency and increased tumour growth compared to A375P cells.

Moreover, using limiting dilution assays we show now that when either *FZD7* or *DAAMI* are depleted from amoeboid-enriched populations in A375M2 or WM1361, tumour initiation decreases (new Fig.5e-l and Supplementary Fig.5h-p). This data further demonstrates the contribution of the amoeboid-rich populations in tumour initiation *in vivo*.

We thank the reviewer for suggesting this critical point.

2) Based on gene expression studies at the population level and on WNT11 kd assays (which affect invasiveness without perturbing proliferation), they conclude that amoeboid cells are invasive and at the same time proliferative, with “no need for phenotype switching”. Again, since the authors don’t resolve the amoeboid state at the single cell level, the claim that cells in an amoeboid state are both invasive and proliferative cannot be made.

Our transcriptional analysis shows that amoeboid A375M2 melanoma cells are enriched in both proliferative and invasive genes when compared with more elongated and less contractile A375P cells. Same is true when comparing A375M2 control with A375M2 treated with either ROCK or Myosin II inhibitors.

Now, we have also investigated:

1. Proliferative and invasive features of melanoma cells *in vivo* in the 4599 orthotopic spontaneous melanoma metastasis model where we recapitulate all steps of melanoma progression (new Fig.6). We show that invading single cells at the distal invasive front (an area of local invasion into the dermis) are all rounded, and remarkably, 30.2% of those rounded invading cells are co-stained with the highest Myosin II score and with ki-67 proliferative marker (new Fig.6c).
2. In the same orthotopic spontaneous melanoma metastasis model, we also observe that cell rounding, overall Myosin II activity and ki-67 positive levels progressively increase from tumour body to invasive front to distal invasive front in control tumours (new Fig.6d-f). ROCK inhibitor (ROCKi) treatment *in vivo* in already established tumours lead to loss of cell rounding and decreased Myosin II activity in the invasive areas, while a reduction of ki-67 positive cells was observed in all areas of ROCKi treated tumours (new Fig.6d-f).
3. We also observe an *in vivo* reduction of cell rounding, Myosin II levels and ki-67 positive cells in tumours derived from A375M2 and B16F10 cells that were pre-treated *in vitro* with ROCKi before injection (without any further treatment of animals) (new Fig.5a-d and Supplementary Fig.5c-g).
4. We also show now an enrichment in rounded cells, high Myosin II levels and ki-67 positive cells in tumour invasive fronts from all our *in vivo* studies (new Fig.1b-e,5e-m and Supplementary Fig.1a,c; 5h-p), while a significant reduction in the ki-67 content is observed when amoeboid features are decreased in A375P compared to A375M2 (new Fig.1b-e and Supplementary Fig.1a,c), and also in either *FZD7* or *DAAMI* depleted tumours compared to shControl tumours in WM1361 and A375M2 cells (new Fig.5e-m and Supplementary Fig.5h-p).
5. In addition, in our serial sphere passages we also find an increase in amoeboid features (cell roundness, high Myosin II activity and blebbing) associated with higher ki-67 proliferative cells (new Fig.1f-k and Supplementary Fig.1e-j).

6. Finally, we now find an enrichment in amoeboid features and ki-67 in the invasive front of primary and metastatic human melanoma tissues (Fig7a,b; 8a,b and Supplementary Fig.7a; 8a).

All these sets of new data suggest that a high percentage amoeboid invading cells are also in a proliferative state.

3) Likewise, the authors propose that ROCK-Myosin II and WNT-FZD-DAAM1 control stemness and invasiveness in amoeboid cells. What's the evidence that these pathways specifically affect cells in an amoeboid state?

We have now assessed the effect of ROCK inhibition and depletion of *FZD7* and *DAAMI* in melanoma cell lines with lower content of amoeboid cells (A375P and WM983A). Similarly to what we show in cell lines enriched for amoeboid states (new Fig.3b,c and Supplementary Fig.3d,e), impaired melanosphere formation and reduced *in vitro* 2D cell viability was found in elongated A375P and WM983A treated with ROCKi (Figure for Reviewer R1.1a,b).

Effects with ROCKi *in vitro* are expected to be stronger than those blocking FZD7-DAAM1, since we saw in amoeboid cells that *-in vitro-* ROCK affects long-term viability while FZD7-DAAM1 doesn't (new Fig.4d,j and Supplementary Fig.4e,k). Nevertheless, note that *in vivo* we measure a reduction of ki-67 proliferative cells in *FZD7* and *DAAMI* depleted cells (new Fig.5 and Supplementary Fig.5).

Accordingly, depletion of either *FZD7* or *DAAMI* in A375P and WM983A cells resulted in decreased melanosphere formation without affecting *in vitro* 2D cell viability (Figure for Reviewer R1.1c,d,g,h). Cell rounding and Myosin II activity were also reduced in *FZD7* and *DAAMI* depleted cells (Fig. for Reviewer R1.1e,f,i,j). Since these cell lines still have a small proportion of amoeboid cells with the FZD7-DAAM1 pathway active, therefore we are affecting such population. Effects observed in these cell lines were less pronounced compared to the ones obtained in cell lines enriched for the amoeboid phenotype (A375M2 and WM1361 cells) (new Fig.4c-f,h-j and Supplementary Fig.4a-g,i-o).

We have not included this set of results in the manuscript as it is already very dense, but if reviewer/editors feel we need to include them we will do so.

Fig. for Reviewer R1.1. Quantification of (a) Sphere Formation Index and (b) cell viability of A375P and WM983A cells treated with one dose of ROCKi (GSK269962A) ($n \geq 2$). (c-j) Quantification of (c,g) Sphere Formation Index ($n \geq 3$), (d,h) cell viability ($n=3$), (e,i) cell morphology (>250 cells pooled from 3 experiments) and (f,j) p-MLC2 immunofluorescence signal relative to the cell area (>80 cells pooled from 3 experiments) of (e-f) A375P and (g-j) WM983A cells after *FZD7* and *DAAM1* knockdown. (a-d,g,h) Graphs show mean \pm SEM (e,f,i,j) Box plots show min to max. P-values generated with (a,b) t-test, (c,d,g,h) One-way ANOVA with Dunnett post-hoc test, (e,f,i,j) Kruskal-Wallis with Dunn's multiple comparison test.

4) For many of their experiments, the authors use only one siRNA (or shRNA) to address the function of a given gene. They have to repeat these experiments with at least two siRNAs/shRNAs per gene.

We have now assessed the effects of *WNT11* depletion in cell morphology, Myosin II levels, sphere formation capacity, cell viability and invasion using three different siRNAs in two cell lines (A375M2 and WM1361) (new Fig.3g,i and Supplementary Fig.2d-g).

Furthermore, the effect of *FZD7* and *DAAM1* silencing on the same *in vitro* features has been investigated using one siRNA and two different shRNAs in two different cell lines (A375M2 and WM1361) (new Fig.4 and Supplementary Fig.4).

In addition, the impact of *FZD7* and *DAAM1* in tumour initiation and amoeboid features has also been addressed *in vivo* in WM1361 and A375M2 cells (new Fig.5 and Supplementary Fig.5).

5) Assessing the tumorigenic capacity of shDAAM1 melanoma cells *in vivo* is one of the key experiments of this study (Fig. 5). This experiment has to be repeated with additional cell lines.

The *in vivo* tumour-initiating capacity of shDAAMI WM1361 melanoma cells has been validated and confirmed using shDAAMI A375M2 melanoma cells (new Supplementary Fig.5l-p).

6) Related to my point 2), proliferation has to be assessed in the experiments using ROCKi (Fig. 2, *in vitro* and *in vivo*) and shDAAM1 (Fig. 5, *in vitro* and *in vivo*). Similarly, proliferation has to be assessed in the *in vitro* experiments presented in Figs. 1b, 1d, and 3h.

We thank the reviewer for this suggestion. We have now assessed 2D cell viability in all *in vitro* experiments (new Fig.1h; 3c,e,f,i,j; 4d,j and Supplementary Fig.1g; 3e; 4e,m; 5a,b). Cell proliferation has also been assessed in all *in vivo* experiments (new Fig.5d,h,l; 6c,f and Supplementary Fig.1c; 5g,o), in melanospheres from our serial sphere passages (new Fig.1h and Supplementary Fig.1g), as well as in the primary and metastatic human patient samples (new Fig.7c; 8c).

Regarding the *in vitro* 2D cell viability of WM983A overexpressing MLC2-GFP (new Supplementary Fig.1.k-o), we observed increased melanosphere formation, cell rounding and Myosin II activity in overexpressing MLC2 WM983A cells compared to control with no effects in 2D cell viability (Fig. for Reviewer R1.2a).

Regarding our screen where we assessed if EMT genes could regulate amoeboid features (new Fig.2d-i), no significant differences in cell numbers were found after depletion of indicated genes (Fig. for Reviewer R1.2b).

Fig. for Reviewer R1.2. (a) Cell viability of WM983A cells over-expressing EGFP-wild type MLC2 or control EGFP. (b) Cell viability after depletion of indicated genes in WM1361 melanoma cells (n=3). Graphs show mean±SEM. P-values generated with (a) t-test and (b) one-way ANOVA with Dunnett post-hoc test.

If reviewer/editors want to include these data we are happy to do so. We have tried to streamline the paper.

7) Non-amoeboid cell lines display increased self-renewal capacity over time, which is associated with acquisition of amoeboid features (Fig. 1). Accordingly, cell lines with amoeboid features (eg A375M2 and others) should have an intrinsically higher self-renewal and tumor-initiation capacity than non-amoeboid cell lines. Is this the case?

Yes, as explained in Point 1 of this Reviewer, using a limiting dilution assay, we now show that amoeboid A375M2 cells have significantly higher tumour-initiating abilities *in vivo* than A375P cells (new Fig.1b-e and Supplementary Fig.1a,c).

8) The role of FZD7 has to be addressed *in vivo* in order to validate the relevance of the WNT11/FZD7/DAAM1 signalling axis (Fig. 4).

We have now examined the *in vivo* tumour-initiating capacity of FZD7 using a limiting dilution assay with shFZD7 WM1361 cells. We demonstrate that FZD7 loss lead to reduced tumour initiation capacity and reduced amoeboid features *in vivo*. Reduced FZD7 levels decrease the fraction of tumour initiating cells and lead to decreased tumour growth at limiting dilutions of 50,000 and 5,000 cells (new Fig.5e). Moreover, reduced: 1) cell rounding, 2) Myosin II activity and 3) number of ki-67 positive cells was observed in FZD7 depleted tumours (new Fig.5f-h and Supplementary Fig.5h,i).

9) The link from the experiments addressing the role of ROCK and WNT11/FZD7/DAAM1 (Figs 1 to 5) to the rather descriptive experiments presented in Figs. 6, 7 is unclear. Importantly, whether cells with high levels of DAAM1, WNT11 also express high ALDH1A1, NANOG, pMLC2, etc, has to be shown at the cellular level. Otherwise, it cannot be stated that “amoeboid cells are linked to poor clinical outcome” (a statement solely based on ALDH1A1 TCGA data; Fig. 6k).

We have observed a decreased in the expression of different stem cell-related markers after both ROCK inhibition (new Fig3a. and Supplementary Fig.3a) and DAAM1 depletion (new Supplementary Fig.4p). We decided to assess further the tumour-initiating amoeboid phenotype sustained by non-canonical Wnt signalling in the clinical setting.

We agree that this set of data is descriptive, since to do single cell analysis “in situ” we would have to do multiplex staining protocol for all the 9 markers, which has proven to be technically challenging. What we provide here is regional information showing how in the same area of tumours all of these markers are highly expressed. However, the stainings were performed in consecutive sections where the histological pattern was always conserved. Furthermore, even with the lack of resolution at cellular level, we do identify these tumour cells at the edge as the major source of detection compared to the rest of tumour microenvironment (immune, stromal and endothelial cells). We now provide regional information for amoeboid cell content (rounded cells with very high Myosin II activity), proliferative marker (ki-67), Wnt markers (DAAM1, WNT11, WNT5B) and stem cell-like markers (ALDH1A1, CD44 and NANOG) in primary and metastatic melanomas from human patients (new Fig.7,8). We also provide for the reviewer hotspot maps for 5 of these markers, really showing that those tumour areas, that can span 1mm diameter, are enriched in cells expressing p-MLC2, ki-67, DAAM1, ALDH1A1 and CD44 markers (Fig. for Reviewer R1.3).

Fig. for Reviewer R1.3. IHC staining and hotspot maps for ALDH1A1, CD44, p-MLC2, DAAM1 and ki-67 stainings in the invasive front and tumour body areas of human primary melanomas.

Regarding prognostic value: we only found prognostic value associated with ALDH1A1 protein expression in the invasive front of primary tumours. The reviewer may have missed that this prognostic value was found analysing protein levels in our cohort of 53 human primary melanoma patients rather than in TCGA mRNA database. Assessing expression in TCGA will not lead to the same observation, as it does not separate TB and IF. That is why we think our study is so important, it highlights the invasive front as a “structure in the tumour with prognostic features”.

We have removed the statement “Overall, these data illustrate that the IFs of human primary melanomas are enriched in amoeboid cells that express non-canonical Wnt and stemness-related genes, and that they are linked to poor clinical outcome” and instead we have stated “Overall, these data illustrate that the IFs of human primary melanomas are tumour areas enriched in amoeboid, proliferative, non-canonical Wnt and stem cell-related markers”.

10) In several figures, the data are presented as bar graphs without showing the actual data points. These should be included, particularly because for many experiments the error bars and

data point distribution seem to be huge. In Fig. 1g, for instance, most data points for relative p-MLC2 levels appear to be very similar in GFP vs MLC2, suggesting that only a minor fraction of data points accounts for the significance in differential p-MLC2 expression. Do the authors have an explanation for the apparently differential responsiveness to MLC2-GFP expression in melanoma cells?

We show now dots representing individual data points in all our Figures.

As explained before in point 6 of this reviewer, we obtained a discrete increase of p-MLC2 after overexpressing MLC2 in WM983A cells since overexpression of GFP-MLC2 leads to loss of endogenous MLC2. The results shown in our previous version of the manuscript (new Supplementary Fig.1o) represented both endogenous and exogenous p-MLC2 levels measured by immunofluorescence. We also provide now for clarity immunoblots where we detect separately endogenous and overexpressed p-MLC2 and MLC2 protein levels (new Supplementary Fig.1k).

11) In Fig. 2, p-MLC2 levels have to be assessed at the end point of the experiments (i.e. 18 days for the A375M2 and 11 days for the B16F10 line).

p-MLC2 levels by IHC as well as cell morphology and ki-67 labelling have now been assessed in both experiments (new Fig.5a-d and Supplementary Fig.5a-g). In both cell lines, reduced cell rounding, p-MLC2 levels and ki-67 positive cells were observed in ROCKi pre-treated tumours. We have now also used two different ROCKi concentrations that show similar results (new Supplementary Fig.5c,d).

12) On p.7, the authors claim that “ROCKi leads to loss of cell rounding and decreased p-MLC2 levels in all areas” of the tumor. However, this doesn’t seem to be the case for the TB area (Fig. 2g, h).

We apologize for this, we meant in all invasive areas. We corrected this in the text.

13) How exactly do the authors determine the melanoma cell shape score and the roundness index? This should be better explained.

We have now better explained this in more detail in the Methods section.

Regarding melanoma cell shape score: Morphologic analysis of tumour tissues was performed on haematoxylin- and eosin-stained (H&E) sections as previously described (Sanz-Moreno, Cancer Cell 2011). Cell shape was graded from 0 to 3 (0=round, 1=ovoid, 2=elongated and 3=spindly) and a cell shape score was assigned to IF and tumour body (TB) regions as follows: cell shape score = ((percentage of cells [%] shape 0 × 0) + (% shape 1 × 1) + (% shape 2 × 2) + (% shape 3 × 3)), with values ranging from 0 (all cells round) to 300 (all cells spindle).

Regarding *in vitro* cell morphology (roundness index): Cell morphology was quantified on phase-contrast images of cells cultured on top of bovine collagen I matrices using ImageJ as previously described (Orgaz et al, Nat Commun 2014; Georgouli et al, Cell 2019; Orgaz et al, Cancer Cell 2020). Cell morphology was assessed using the morphology descriptor tool “roundness” after manually drawing around the cell shape. Roundness index is calculated as 4

$\times \text{area} / \pi \times \text{major_axis_length}^2$. Values closer to 1 represent rounded morphology; values closer to 0 represent more spread and/or spindle-shaped cells.

14) In Fig. 2o, decreased metastasis of ROCKi-treated tumors is interpreted as decreased invasiveness/metastatic capacity. However, decreased numbers of metastases at distant sites might simply be due to the decreased size of the primary tumors.

As we could not separate the effects of ROCKi in primary tumour growth to the impact on metastasis in our orthotopic spontaneous melanoma metastasis model, we have now evaluated the impact of ROCKi in metastatic colonisation and outgrowth using tail vein injection assays (new Fig.6). 4599 cells were pre-treated with ROCKi for 24 hours prior to intravenous injection and continued treatment *in vivo* for 6 days. Cell survival in the blood stream or cell extravasation was not altered as similar cell numbers lodged in the lungs 30 minutes post-tail vein injection (new Supplementary Fig.6e). However, ROCKi significantly reduced lung metastatic establishment (new Fig.6m). Furthermore, lower lung metastasis competence was also observed when 4599 cells (without any pre-treatment) were intravenously injected, cells lodged in the lung and 4 hours after, mice received ROCKi systemic treatment for 12 days (new Fig.6n).

These combined datasets show that ROCK-Myosin II activity is not only important for tumour formation and invasion but also for early colonization and metastatic growth.

15) Do the authors have any explanation for how ROCK might control EMT-related genes, including WNT11? This should at least be discussed.

This has been now included in the discussion of the manuscript.

Rho-ROCK signalling has been shown in other models to regulate actin cytoskeleton rearrangement during EMT (Lamouille et al, Nat Rev Mol Cell Biol 2014), while promoting transcriptional changes through activating p38 signalling (Zhou et al, Cancers 2019; Chen et al, Exp Clin Cancer Res 2018). Interestingly, ROCK regulates TGF- β expression in our EMT array and TGF- β signalling was one of the top networks associated with the EMT gene signature regulated by ROCK. We have also shown that amoeboid cells secrete high levels of TGF β and its secretion is ROCK dependent (Cantelli et al, Curr Biol 2015; Georgouli et al, Cell 2019). TGF β is a key driver of EMT by inducing transcription of several mesenchymal genes and increasing the activity of EMT transcription factors (Lamouille et al, Nat Rev Mol Cell Biol 2014; Cantelli et al, Semin Cancer Biol 2017). In agreement with this, we find that in amoeboid cells that secrete high levels of TGF β , ROCK inhibition leads to significantly reduced expression of key EMT regulators, such as SNAI family members and the mesenchymal markers N-cadherin, Vimentin and Fibronectin (new Fig.2c). These observations suggest that TGF- β signalling is one of the possible mechanisms by which ROCK modulates a global EMT program and amoeboid invasion (Cantelli et al, Curr Biol 2015).

Rev#2 Expert in melanoma and stem cells

Melanoma stemness is fueled by Wnt-DAAM1 signaling in the amoeboid invasive compartment

The authors herein report interesting findings revealing the role of amoeboid melanoma cells in stemness and metastasis. The authors identify a Wnt-DAAM1 axis as critical for the aggressive phenotypes of amoeboid cells, providing rationale to therapeutically target this population of melanoma cells. The findings will be of particular interest to the melanoma field, however the below issues will need to be addressed before publication:

- The abstract needs rewriting for clarity

It has been rewritten in a clearer way and including new findings.

- Figure 1B: the authors use A375P and WM983A because they have been previously reported to have low levels of Myosin II.

o The authors should here repeat WB or RT-PCR of Myosin II in these cell lines to demonstrate how robust this is.

As previously reported (Orgaz et al, Nat Commun 2014 Fig.1c; Herraiz et al, J Natl Cancer Inst 2015 Fig.1c; Georgouli et al, Cell 2019 Fig2a,g; Orgaz et al, Cancer Cell 2020 Supplementary Fig1.g), we now include immunoblots for A375P vs A375M2, and WM983A vs WM983B to show that A375P and WM983A have lower levels of Myosin II activity (new Supplementary Fig.1d).

o Two cell lines that have high Myosin II levels should also be used to demonstrate the importance of Myosin II levels in the discussed phenotypes

Serial sphere passages of A375M2 and WM983B amoeboid melanoma cells with high Myosin II levels have now been performed. As expected, the enrichments were less pronounced than in elongated low Myosin II cells. Nevertheless, we found increased melanosphere formation capacity associated with more rounded-amoeboid cells expressing high levels of Myosin II, higher blebbing and higher ki-67 positive cells (new Supplementary Fig1e-j). Interestingly, in already very rounded cells an increase in p-MLC2 translates in increased membrane blebbing (new Supplementary Fig.1j).

Moreover, as explained to Reviewer#1, the tumour-initiating capacity of A375M2 and A375P cells and the impact of the amoeboid phenotype has also been addressed *in vivo* using limiting dilution assays (new Fig.1b-e and Supplementary Fig.1a,c). Our new *in vivo* data shows how A375M2 have higher tumour-initiating potential than A375P.

- The authors should probe for amoeboid markers by WB or RT-PCR following the serial passaging in low adherent conditions (Figure 1B) and collagen I (Figure 1C) to strengthen their scientific point.

To address this point, we now provide:

1. Myosin II activity (p-MLC2 levels) has now been assessed *in situ* in the spheres by immunohistochemistry (new Fig.1g and Supplementary Fig.1f). We find a progressive increase in Myosin II activity with serial passaging.
2. Furthermore, we also analysed p-MLC2 by immunofluorescence at single cell level after dissociation of spheres, and seeding on collagen I matrices (new Fig.1j and Supplementary Fig.1i) showing how single cells have more Myosin activity II with passaging.
3. In addition, as a feature of “amoeboidness”, number of blebbing cells was also quantified at the single cell level (new Fig1k and Supplementary Fig.1j). Increased number of blebbing cells is observed with increasing passage number.

In all cases we observe increased amoeboid markers/features with serial passaging.

- For Figure 1G, show the WB for the phospho-MLC2 levels for the reader to appreciate

In the previous version of the manuscript, p-MLC2 levels for that experiment were assessed by immunofluorescence (new Supplementary Fig.1o). These results showed both endogenous and exogenous p-MLC2 levels. We also provide now for clarity immunoblots where we detect separately endogenous and overexpressed p-MLC2 and MLC2 protein levels (new Supplementary Fig.1k). As it can be seen in the blots exogenous GFP-MLC2 over expression results in a decrease in the endogenous protein.

- Why do the authors use an NRAS-MT cell line (WM1361) in Figure 1K. What is the justification for which genotype of melanoma to focus on?

Since the main genetic alterations in melanoma are BRAF^{V600E} and NRAS^{Q61L}, we used WM1361 cell line as an example of an NRAS^{Q61L} amoeboid melanoma line to prove that our observations are oncogene independent. We have now extended most of our observations to both genetic backgrounds *in vitro* (new Fig.2b-h; 3b-j; 4c-k and Supplementary Fig.2d-j; 3d-g; 4a-g,i-o) and *in vivo* (new Fig.5e-l and Supplementary Fig.5h-p) using both A375M2 (BRAF^{V600E}) and WM1361 (NRAS^{Q61L}) (as they hold true both for BRAF^{V600E} and NRAS^{Q61L} driven melanoma) to assess the WNT11/5B-FZD7-DAAM1-ROCK-Myosin II axis in all our assays.

- What are the impacts on melanoma viability from using ROCK inhibitors and silencing ROCK and MLC2 genes?

We have now included the effects of ROCKi and *ROCK* and *MLC2* gene silencing in 2D cell viability *in vitro*. We have analysed several amoeboid melanoma cell lines (new Fig.3c,e,f and Supplementary Fig.3e). *In vitro* 2D cell viability was reduced after 7 days (Fig.3c,e and Supplementary Fig.3e), while no effects in 2D cell viability were measured after 3 days (new Fig.3f), suggesting that we can transiently reduce Myosin II levels without affecting cell viability but long term-term depletion compromises cell survival. For this reason, we further show how ROCK-Myosin II sustains self-renewal via regulation of Wnt ligands, but Wnt signalling does not compromise cell numbers *in vitro*. So, *in vitro*, ROCK regulates both 2D viability and self-renewal but non-canonical Wnt only regulates the latter.

Nevertheless, note that *in vivo* we measure a reduction of ki-67 proliferative cells after ROCKi treatment and also in *FZD7* and *DAAMI* depleted cells.

- How were A375M2 cells generated? The authors unfortunately use only 1 amoeboid cell line for many experiments and it is difficult to appreciate how robust these findings would be in an expanded cell line panel.

Amoeboid A375M2 cells were specifically derived and isolated from A375P cells (which are isolated from a lymph node metastasis) after two rounds of lung metastatic colonization (Clark et al., Nature 2000), therefore A375M2 are considered “super” metastatic. As mentioned above, we have now included more amoeboid melanoma cells lines (WM1361 and WM983B) in our *in vitro* sphere assays using ROCKi and depleting *ROCK* and *MLC2* genes (new Fig.3b,d and Supplementary Fig.3d). WM983B/A are isogenic lines too: WM983B (metastatic, high Myosin II and amoeboid) was isolated from the same patient as WM983A (primary tumour, low Myosin II, elongated) (Georgouli et al, Cell 2019 Orgaz et al, Cell 2020).

- Figure 2, A375M2 cells were treated for 5 days before *in vivo* transplantation with ROCKi. What is the impact on melanoma cell viability after this treatment?

In those experiments using A375M2 and B16F10 melanoma cells, 5 days pre-treatment with ROCKi reduced 2D cell viability (new Supplementary Fig.5a,b). We have also used now a lower concentration of ROCKi (new Supplementary Fig.5b-d) and found similar results. However, after this pre-treatment, the same amount of viable cells (assessed by Trypan blue) were subcutaneously injected into mice (new Fig.5a and Supplementary Fig.5c). We also assessed ki-67 staining in all tumours at the end point of the experiments. A reduction in the fraction of ki-67 positive cells was also observed in all cases (new Fig.5d and Supplementary Fig.5g). For this reason, we have emphasized the specific role of WNT11-FZD7-DAAM1 axis crosstalk with ROCK in tumour-initiating abilities/amoeboid invasion rather than global ROCK signalling.

- o If ROCKi causes extensive cytotoxicity *in vitro*, obviously an impact on *in vivo* growth would be observed

As mentioned above, long-term (but not short-term) inhibition of ROCK-Myosin II has an impact on cell viability. This is the reason we further investigated the specific role of non-canonical Wnt pathway in regulating tumour initiation independently of regulating cell viability. We use ROCK inhibitor in some experiments *in vivo* for 2 reasons: to abolish the amoeboid phenotype and because to our knowledge there are no inhibitors to block non-canonical pathway for *in vivo* work. Several compounds target canonical Wnt/ β -catenin pathway and some of them such as PORCN inhibitors can shut down both canonical and non-canonical Wnt signalling, but *in vivo* drug inhibition of solely non-canonical Wnt signalling is not possible (Zhong et al, Mol Pharmacol 2020; Zimmerli et al, Br J Pharmacol 2017). Moreover, our work aims to dissociate the specific WNT11-FZD7-DAAM1 response from a more global ROCK response.

- Figure 2O,P: It is difficult to determine whether the ROCK inhibitor decreases spontaneous metastases or the ROCKi pretreatment before injection into mice kills all melanoma cells outright.

In the *in vivo* orthotopic spontaneous melanoma metastasis model that the reviewer is referring to, 4599 melanoma cells were not pretreated before injection (now new Fig.6a-k). Treatment started after establishment of tumours (day 11: tumours are palpable). Reviewer must not get confused with experiments performed in A375M2 and B16F10 cells for assessing tumour initiation (new Fig.5a-d and Supplementary Fig.5a-g), in which cells were pre-treated with ROCKi, but then never treated again *in vivo*. These are two very different settings.

Furthermore, for assessing the role of ROCKi in metastasis, we have now performed tail vein injection assays to evaluate the impact of ROCK in melanoma metastatic colonisation and outgrowth independently of primary tumour growth. 4599 cells were either pre-treated with ROCKi for 24 hours prior to intravenous injection and continued treatment *in vivo* for 6 days, or 4599 cells without any pre-treatment were intravenously injected and mice received ROCKi systemic treatment for 12 days (new Fig.6m,n). In all cases we obtained similar reduction in metastasis formation.

- Figure 3C needs statistics. Which of these changes in gene expression are significant?

We now show 4 independent biological replicates of the EMT qPCR array and significant statistical differences were calculated (new Fig.2c).

- Figure 3E: It is confusing why there are also stars indicating statistical significance achieved in cells treated with no (-) siRNA?

We apologize for that, it was a formatting error. It has already been corrected (new Fig.2e).

- o There does not appear to be a significant difference between any condition relative to the (-) control. The same goes for Figure 3F. The authors should show every replicate as a dot.

We show now dots representing individual data points for all graphs in the new version of the manuscript and a significant difference.

- Figure 4 C, Show every measurement as an individual dot. It is difficult to see how the data shows statistical differences by the way the data is currently displayed.

We show now individual dots for all graphs in the new version of the manuscript.

- Are the data shown in 4D and 4E showing only statistically significant changes? Need statistics.

The heat maps included the fold change in expression for non-canonical Wnt receptors and co-receptors. We have now included all statistical analysis (new Fig.4a,b).

Reviewer #3 (Remarks to the Author); expert in cell migration and morphodynamics:

Rodriguez-Hernandez and coauthors present extensive evidence from *in vitro* experiments, several *in vivo* models, patient tissue samples, and previously published datasets to describe a novel pathway in melanoma whereby non-canonical soluble Wnts (Wnt11 and Wnt5b) induce myosin II activity through an FZD7-DAAM1-ROCK signaling axis. In cell culture experiments with melanoma cell lines, heightened myosin activity as evidenced by phosphorylated MLC (pMLC) was associated with a rounded cell morphology reminiscent of cells migrating in an amoeboid fashion, higher capacity to form melanospheres, and expression of genes associated with stemness and EMT. Melanoma cells transplanted into mice formed tumors with rounded cells at the tumor margin and showed potential for metastasis to the lung, but pre-treatment with a ROCK inhibitor diminished these features. The authors used siRNA/shRNA to flesh out the pathway showing that autocrine or paracrine Wnt ligands engage the FZD7 receptor, signal through Dvl-DAAM1-Rho, and induce downstream myosin activity associated with a more rounded cell morphology. Having established these key components and after implicating this pathway in tumor growth and metastasis *in vivo*, they then histologically examine human primary and metastatic melanoma tissue for markers of stemness, non-canonical Wnt markers, and amoeboid markers (pMLC and shape) showing that in both locations these markers are heightened at the tumor margin or invasive front. In sum, the authors put forth that an invasive front enriched with amoeboid cells with expression profiles indicative of both stemness and EMT are a key reason for the aggressiveness of malignant melanomas and a potential prognostic indicator. This body of work implying a simultaneous migratory and stem-like cancer cell phenotype is deemed novel given current notions of cancer stem cell function and location within the tumor and previous work from Friedl and colleagues on the topic of amoeboid-mesenchymal transition. The work provides new insights and is extensive, and merits publication in Nature Communications, but a few concerns are listed below:

Major:

1. A major implication of the work is that cells with an amoeboid phenotype are linked to the aggressiveness of melanoma. The term amoeboid is a descriptor of a cell's migratory mode and not just shape – conventionally involving proteolytic-independent migration with limited adhesion to the ECM, high cytoskeletal contractility, cytoskeletal extensions via short protrusions or blebs, and a rounded (unspread) morphology. The authors heavily rely on this last descriptor (rounded cell morphology) to implicate amoeboid migration in their findings, but do not directly assess migration phenotype. Assays for attachment to keratinocytes and invasion into collagen matrices were performed (Figure 3), but neither of these directly supports a central role for amoeboid migration in the presented studies. While this is obviously quite difficult to obtain from *in vivo* studies, *in vitro* studies (either timelapse or fixed time-point as in Figure 3h demonstrating invasion in the presence of an MMP inhibitor) would strengthen this link.

First, we would like to clarify that in this work, we have used models of very well established melanoma amoeboid invasion since we have already previously extensively characterised using *in vivo* intravital imaging (Sanz-Moreno et al, Cell 2008; Sanz-Moreno et

al, Cancer Cell 2011; Herraiz et al, J Natl Cancer Inst 2015), transwell migration assays (Georgouli et al, Cell 2019), 3D invasion assays (Sanz Moreno et al, Cell 2008; Sanz-Moreno, Cancer Cell 2011; Orgaz et al, Nat Commun 2014; Cantelli et al, Curr Biol 2015), and cell migration tracking using video lapse (Sanz-Moreno, Cell 2008; Sanz-Moreno, Cancer Cell 2011; Orgaz et al, Nat Commun 2014; Herraiz et al, J Natl Cancer Inst 2015).

The current study wants to study further our understanding of the “amoeboid state” is not just a migratory mode but a tumour-initiating cell population. Nevertheless, we have now assessed more features to better characterize the amoeboid phenotype after manipulation of non-canonical Wnt signalling and ROCK-Myosin II pathways:

1. In the original version, we had already provided analysis of cell morphology and p-MLC2 levels in all our *in vitro* and *in vivo* experiments. We have also analysed now the percentage of blebbing cells in individual dissociated cells from serial passaged spheres (new Fig.1i-k and Supplementary Fig.1h-j).
2. 3D invasion assays through collagen I matrices have now been performed in *WNT11*, *FZD7* and *DAAMI* depleted amoeboid melanoma cells (new Fig.4g,k and Supplementary Fig.2g) in addition to our initial screen (new Fig.2h) that was already present in the original manuscript.
3. Moreover, although it is also difficult to measure the amoeboid phenotype *in vivo*, in all our *in vivo* experiments we have analysed cell morphology and phosphorylated Myosin II levels (new Fig1,5,6) as well as in primary and metastatic melanoma lesions from human patients (new Fig.7,8).
4. Importantly, in the orthotopic spontaneous melanoma metastasis model where we recapitulated the steps of melanoma progression (Fig.6), we can identify where cells are locally invading the dermis (*in vivo*) and they all have amoeboid features. In fact, all invading cells at the distal invasive front have a rounded morphology: 56% of them exhibit the highest Myosin II activity (score 3) and 60.4% of invading cells are positive for the proliferative marker ki-67 (Fig.6c). Remarkably, co-staining of p-MLC2 score 3 and ki-67 was found in 30.2% of those invading cells.
5. On the other hand, it has been proposed that amoeboid cancer cells do not require MMPs to invade (Sahai et al, Nat Cell Biol 2003; Wolf et al, J Cell Biol 2003; Wyckoff et al, Curr Biol 2006). However, we have previously observed that amoeboid melanoma cells secrete higher levels of several MMPs (Orgaz et al, Nat Commun 2014; Georgouli et al, Cell 2019), including MMP9 and MMP13, and are capable of degrading matrigel and collagen I respectively to some extent (Orgaz et al, Nat Commun 2014). As such, following the reviewer suggestion, we assessed 3D invasion of amoeboid A375M2 cells through collagen I and mixed Matrigel/collagen I (1:1) matrices in the presence of MMP9 inhibitor and the broad spectrum MMP inhibitor GM6001. MMP-9 inhibitor only reduces invasion of A375M2 cells into Matrigel/collagen I, while the GM6001 decreases invasion in both collagen and Matrigel/collagen I matrices (Fig. for Reviewer R3.1a,b), but there is still some remaining invasion. This data indicates that there is some level of dependence on MMPs in these amoeboid cells, but some amoeboid invasion can still take place in the absence of MMP activity-as expected.

Fig. for Reviewer R3.1. Quantification of 3D invasion index into a (a) collagen I or (b) Matrigel/collagen I (1:1) matrix of A375M2 treated with MMP9 inhibitor (1 μ M) or GM6001 inhibitor (25 μ M) (n \geq 3). Graphs show mean \pm SEM. P-values generated with one-way ANOVA with Dunnett post-hoc test.

Since amoeboid melanoma cells depend -to some extent- on MMPs, we did not think it was necessary to perform these assays interfering with non-canonical Wnt as it would give rise to confusing results.

2. In emphasizing the involvement of amoeboid migration, the authors perform experiments to show that the proliferative capacity of these cells is unaffected by the Wnt-FZD7-DAAMI-ROCK-MLC pathway. They seem to imply that the amoeboid migratory phenotype is a driver of tumor growth and spread, de-emphasizing the role of proliferation which presumably is critical to melanosphere formation and primary tumor growth. Given many previous connections between myosin activity, cytoskeletal contractility, and proliferation, one would expect the ROCK inhibited cells in the studies presented in Figure 2 to proliferate less. Related to this, the quantification of proliferation isn't clear: as described in the Methods, crystal violet was used to count cells after 3-7 days and this value was normalized to the value of initially plated cells. All of the provided quantifications are at or near 100% - does this imply cells did not proliferate over this period (which should be specified more precisely)? The authors should clarify in the text the role of proliferation (perhaps de-emphasizing amoeboid migration given comment #1 above). Additionally, including a better assessment of proliferation such as EdU incorporation or Ki67 staining performed in their sphere formation assays (eg. in comparing A375P vs A375M2, or with contractility perturbations) would provide a clearer sense of how proliferation is altered and potentially contributing to in vivo outcomes.

1. We have now assessed 2D cell viability in all *in vitro* experiments (new Fig.1h; 3c,e,f,i,j; 4d,j and Supplementary Fig.1g; 3e; 4e,m; 5a,b) and further investigated proliferation by ki-67 staining in all *in vivo* experiments with ROCKi, and stable shFZD7 and shDAAMI melanoma cells (new Fig.5d,h,l; 6c,f and Supplementary Fig.1c; 5g,o). Moreover, ki-67 staining was also performed in primary and metastatic human patient samples (new Fig.7c; 8c).
2. In addition, as suggested by the reviewer, we have further investigated ki-67 proliferative population by immunohistochemistry in paraffin-embedded spheres from serial sphere passages (new Fig.1k and Supplementary Fig.1g). Increase in ki-67 positive cells is observed with increasing passage number.

3. Moreover, we have now performed a limiting dilution assay to assess the *in vivo* tumour initiation capacity of A375M2 cells compared with more elongated A375P cells (new Fig.1b-e and Supplementary Fig.1a-c) where we also investigated both amoeboid and proliferative features in the tumours. We can conclude that amoeboid cells with intrinsically high Myosin II activity are also proliferative and promote tumour growth.
4. Moreover, the focus of the paper was not ROCK but non-canonical Wnt signalling. We use ROCK inhibitor in some experiments *in vivo* for 2 reasons: to reduce the amoeboid phenotype and because to our knowledge there are no inhibitors to block WNT11-FZD7-DAAM1 pathway for *in vivo* work. Several compounds target canonical Wnt/ β -catenin pathway and some of them such as PORCN inhibitors can shut down both canonical and non-canonical Wnt signalling, but *in vivo* drug inhibition of solely non-canonical Wnt signalling is not possible (Zhong et al, Mol Pharmacol 2020; Zimmerli et al, Br J Pharmacol 2017). Moreover, our work aims to dissociate the specific WNT11-FZD7-DAAM1 response from a more global ROCK response.
5. We apologise for not explaining in a clearer way the assessment of cell proliferation. *In vitro*, we have analysed 2D cell viability using Crystal violet assay while in our *in vivo* experiments we have assessed ki67 staining by immunohistochemistry. We have now improved our explanation on quantifying cell viability *in vitro* in the Methods section: Cells stably expressing shRNAs or cells after 48 h transfection with siRNAs were seeded in 6-well plates (25,000 cells/well). For ROCKi experiments, 25,000 cells were seeded in 6-well plates, allowed to adhere and treatment was performed as indicated for each experiment. Cell viability was assessed after 7 days in all experiments, while for *MYL9* and *MYL12B* siRNA experiments that it was assessed 3 and 7 days after seeding. Then, cells were fixed with 1% formaldehyde and stained with 0.25% crystal violet. After air-drying, crystal violet stain was dissolved in 10% acetic acid. Samples from each well were transferred to a 96-well plate and the absorbance was measured at 595 nm on a microplate reader as an indirect measure of cell number. Results were normalized to the value of initially plated cells and presented as percentage of cells in each sample relative to control. This data indicates is that after 3 days in the viability assay (5 days after transfection) we see no difference in viability after suppressing contractility (and this is how most invasion and phenotypic assays have been done in all our past work). Nevertheless, viability in long term assays (7 days in the assay but 9 days after transfection) is measuring the balance between proliferation and death, and this is affected by reducing contractility.

3. Can the authors comment on how they can rule out the influence of ROCK inhibition on initial engraftment in *in vivo* studies? Given the role of myosin/contractility in adhesion, would could imagine that ROCK inhibition leads to poor initial attachment and thus significant cell death immediately following injection..

As explained to Reviewer 2, after 5 days treatment with ROCKi, 2D cell viability of A375M2 and B16F10 melanoma cells was investigated (new Supplementary Fig.5a,b) and same numbers of viable cells assessed (by Trypan blue) were further subcutaneously injected into mice (new Fig.5a and Supplementary Fig.5c).

We have further analysed the impact on cell adhesion to Matrigel/collagen I (1:1) matrices after pre-treatment in those viable cells (Fig. for Reviewer R3.2.a,b). Matrigel matrix was also used for suspension of cells for subcutaneous injection. However, no significant differences were observed in cell adhesion.

Fig. for Reviewer R3.2. After 5 days pre-treatment with ROCKi, quantification of (a) A375M2 and (b) B16F10 adhered cells after 30 minutes, 1 hour and 2 hours of seeding on a Matrigel/collagen I (1:1) matrix (n=4). Graphs show mean±SEM. P-values generated with two-way ANOVA with Dunnett post-hoc test.

o If ROCKi causes extensive cytotoxicity *in vitro*, obviously an impact on *in vivo* growth would be observed

As mention above, ROCKi has an impact on long-term cell viability. This is the reason we further investigated the specific role of non-canonical Wnt pathway in regulating self-renewal. Since ROCK is in the same pathway but has multiple functions one of the aims was really to dissociate the specific WNT11-FZD7-DAAM1 response from a more global ROCK response.

Minor:

1. The title could confuse many readers as “amoeboid invasive compartment” is not an established term and what the compartment is or where it’s location is are both ambiguous. Additionally, given comment #1 above, if the mode of cell migration is not shown to amoeboid, a title change would be warranted.

We have now changed the title from “Melanoma stemness is fuelled by Wnt-DAAM1 signalling in the amoeboid invasive compartment” to “WNT11-FZD7-DAAM1 signalling supports tumour initiating abilities and melanoma amoeboid invasion”.

If the reviewer feels that this title should be further modified, we would be happy to do so.

2. The quantification for cell shape *in vitro* is not sufficiently described, despite the provided range of 0 to 1. Including an equation would be helpful. Related to this, showing cell outlines (of ~10-20 cells) throughout would be more convincing than fluorescent images of only 1-3 cells.

We have better explain the quantification of cell shape *in vitro* in the Methods section: Cell morphology was quantified on phase-contrast images of cells cultured on top of bovine collagen I matrices using ImageJ as previously described (Orgaz et al, Nat Commun 2014; Georgouli et al, Cell 2019; Orgaz et al, Cancer Cell 2020). Cell morphology was assessed using

the morphology descriptor tool “roundness” after manually drawing around the cell shape. Roundness index is calculated as $4 \times \text{area} / \pi \times \text{major_axis_length}^2$. Values closer to 1 represent rounded morphology; values closer to 0 represent more spread and/or spindle-shaped cells.

Specific number of cells quantified and number of experiments performed are indicated in the Figure legend for each experiment. In all our experiments, approximately 80-100 cells were quantified per experiment, and at least three independent biological replicates were performed to assess cell morphology.

We have also provided now better images showing larger number of cells.

3. Labeling of controls as “-“ in Figure 3b westerns is slightly confusing.

The labelling has changed to “control” for clarification.

4. Figure 3e “****” above control – presumably the indicated significances above other groups are comparisons to the control?

We apologize for that, it was a formatting error. It has already been corrected (new Fig.2e).

5. Do the pMLC2 results change when local variations in cell density are accounted for? An unspread, rounded cell morphology would generally lead to higher density of any stained cytosolic protein compared to spread cells that are additionally more separated from each other.

In the experiments were p-MLC2 was quantified by immunofluorescence, cell density did not play a role as we always seed similar number of cells for these experiments (4,000 per 96 well plate) and always the cells were in low confluence conditions. p-MLC2 signal of one single confocal plane was normalized by cell area. Active myosin II has a more diffuse localisation throughout elongated cells while it is organised as a thick cortex in amoeboid cells as shown in new Fig.1j.

For Western blot experiments, total protein from the same number of cells for each condition was extracted, and p-MLC2 levels were normalized versus total MLC2 protein in the lysate.

6. The experiment presented in Figure 2m-p is not sufficiently described in the text or figure legend.

This experiment has been better explained and further extended with tail vein assays (new Fig.6k-m).

We thank all reviewers for their very useful comments as we think their suggestions have greatly improved our work.

REVIEWER COMMENTS

Reviewer #1 (Remarks to the Author):

For the revision of their study, the authors have meticulously addressed my concerns, having performed a series of important additional experiments. The extensive new data are convincing and strongly support the authors' conclusions. Overall, the paper has been greatly improved and is, in my view, ready for publication.

I prefer the new title.

If possible, I recommend incorporation of "Fig. for Reviewer R1.3" as a Supplementary Figure. The data are nice and truly help to better understand how the authors have quantified and interpreted the IHC data.

Reviewer #2 (Remarks to the Author):

The authors have done a tremendous job with this revision. The manuscript addresses observations concerning amoeboid melanoma cells that have neural crest-like features which may underlie their elevated metastatic capacity. The authors provide novel insights and vulnerabilities of this subpopulation and the manuscript will be of great interest across the melanoma field as well as stem cell and EMT fields. All of my prior concerns have been adequately addressed with the additional experiments.

Below are just two minor points for clarification.

- Figure 1B: The difference in tumor-initiating frequency does not appear to be robust, and does not appear 5-fold different as the authors claim. For example, looking at the 5,000 cells injected row; 12 tumors grew from 17 injections for the A375M2, which is almost the same frequency seen in the A375P where 12 tumors grew from 18 injections. Similar results are seen in the other cell dilutions with the difference being 1 tumor, making the 5-fold difference claim unclear. There does not appear to be a significant difference in tumor initiation
- Figure 1C: it is unclear to me why in Figure 1B it is reported that only 8 tumors formed from 10 injections for A375P when 50,000 cells were injected, but then Figure 1C shows tumor volume from 10 tumors, suggesting every single injection of 50,000 A375P cells produced a tumor (10 out of 10). Can the authors clarify?

Reviewed by Vito W. Rebecca, PhD

Reviewer #3 (Remarks to the Author):

The resubmission from Sanz-Moreno and coauthors considerably strengthens the original submission. The authors have added numerous additional experiments appropriately addressing most of the reviewers' legitimate concerns. They have clarified several key methods that raised questions, addressed questions regarding the influence of proliferation and viability, and made appropriate references to their previous work characterizing the migratory behavior of the amoeboid cells reported here which was a major concern for this reviewer. In short, the rebuttal and associated revised manuscript are thoroughly and thoughtfully constructed, and the authors should be commended for their efforts. The finding is novel, well-supported by the experiments, and will be of interest to several audiences. No further revisions are recommended.

REVIEWER COMMENTS

Reviewer #1 (Remarks to the Author):

For the revision of their study, the authors have meticulously addressed my concerns, having performed a series of important additional experiments. The extensive new data are convincing and strongly support the authors' conclusions. Overall, the paper has been greatly improved and is, in my view, ready for publication.

I prefer the new title.

If possible, I recommend incorporation of "Fig. for Reviewer R1.3" as a Supplementary Figure. The data are nice and truly help to better understand how the authors have quantified and interpreted the IHC data.

We are glad that the Reviewer is satisfied with our revised manuscript, we believe our work has greatly improved after addressing them. We have now incorporated "Fig. for Reviewer R1.3" as a Supplementary Fig.7e.

Reviewer #2 (Remarks to the Author):

The authors have done a tremendous job with this revision. The manuscript addresses observations concerning amoeboid melanoma cells that have neural crest-like features which may underlie their elevated metastatic capacity. The authors provide novel insights and vulnerabilities of this subpopulation and the manuscript will be of great interest across the melanoma field as well as stem cell and EMT fields. All of my prior concerns have been adequately addressed with the additional experiments.

Below are just two minor points for clarification.

We thank the reviewer for the comments on our revised manuscript. We have addressed these two minor points.

- Figure 1B: The difference in tumor-initiating frequency does not appear to be robust, and does not appear 5-fold different as the authors claim. For example, looking at the 5,000 cells injected row; 12 tumors grew from 17 injections for the A375M2, which is almost the same frequency seen in the A375P where 12 tumors grew from 18 injections. Similar results are seen in the other cell dilutions with the difference being 1 tumor, making the 5-fold difference claim unclear. There does not appear to be a significant difference in tumor initiation

The fold difference reported in Fig.1b was related with the differences observed in tumour-initiating frequency (TIF) in A375M2 (1/8406) compared to A375P (1/40984). TIF was determined using extreme limiting dilution analysis (ELDA) as set up by Hu and Smyth, *J Immunol Methods* 2009. This is the established method to analyse this kind of experiments and it has been used in many important papers in the field (Perry et al, *Nat Cell Biol*, 2020; Xie et al, *Cell Stem Cell*, 2019; Pastushenko et al, *Nature*, 2018; Pascual et al, *Nature*, 2017; de Sousa e Melo, *Nature* 2017; Piskounova et al, *Nature*, 2015; Calon et al, *Nature Genetics*, 2015; Boumahdi et al, *Nature* 2014; Kreso et al, *Nat Medicine*, 2014; Guo et al, *Cell*, 2012; Ishizawa et al, *Cell Stem Cell*, 2010). We have explained this better in the manuscript. Such

differences later translate in a significant difference in tumour volumes at all dilutions (Fig. 1c).

ELDA analysis is implemented in ELDA webtool software (<http://bioinf.wehi.edu.au/software/elda/>) that is especially suitable for analysing limiting dilution data arising in stem cell research. ELDA tool is designed based on single-hit Poisson model. ELDA considers the number of positive events in each dose of cells injected to estimate the TIF and confidence intervals in each group and to compute differences in TIF between groups using the statistical method of maximum likelihood estimation (Hu and Smyth, J Immunol Methods, 2009).

Although the differences in the frequency of positive tumours between A375M2 and A375P in each cell dilution are not large, they are enough to lead significant differences in the estimated TIF for each cell line when ELDA analysis is applied (Fig. 1b and Fig. for Reviewer R2.1). As shown in Fig. for Reviewer R2.1, for ELDA analysis, the log proportion of negative tumours (“log fraction nonresponding” in the Y-axis) and the dose of cells injected (X-axis) are plotted. For example, looking at the 50,000 cells condition in Fig.1b, we have 9 of 10 positive tumours for the A375M2 that means a negative fraction of 1/10 and a log value of -2.30. While 8 of 10 positive tumours for A375P means a negative fraction of 2/10 and a log value of -1.60. Then, the slope of the line for all dilutions is used by ELDA software to estimate the TIF.

Fig. for Reviewer R2.1. Plot showing the log-fraction of the limiting dilution model fitted to the A375M2 and A375P data from the Table of Fig.1b. The slope of the line is used to calculate the TIF by ELDA and the dotted lines represent the 95% confidence intervals.

If reviewer/editors think these graphs would be more informative than the tables we have in the manuscript for this experiment and for the *in vivo* limiting dilution experiments with WM1361 and A375M2 stable cell lines (Fig.5e, 5i, Supplementary Fig.5l) we can include the graphs instead (Fig. for Reviewer R2.2.).

Fig. for Reviewer R2.2. Plot showing the log-fraction of the limiting dilution model fitted to the data from the Tables of Fig.5e, 5i and Supplementary Fig.5l. The slope of the line is used to calculate the TIF by ELDA and the dotted lines represent the 95% confidence intervals.

- Figure 1C: it is unclear to me why in Figure 1B it is reported that only 8 tumors formed from 10 injections for A375P when 50,000 cells were injected, but then Figure 1C shows tumor volume from 10 tumors, suggesting every single injection of 50,000 A375P cells produced a tumor (10 out of 10). Can the authors clarify?

Reviewed by Vito W. Rebecca, PhD

We apologize for not explaining this more clearly. In Fig.1c we show the tumour volume for all injections in the experiment, while to estimate the TIF using ELDA in Fig. 1b, the bottom 20th percentile of tumour size in the global experiment was used as a threshold for positive tumour initiation (approximately 10 - 20 mm³). We have now explained this in the Methods section. Those tumours that were too small to be considered for ELDA were mostly composed of Matrigel from the injection.

We have now removed from the immunohistochemical analysis in all the *in vivo* limiting dilutions assays included in the manuscript (Fig.1, Fig.5 and Supplementary Fig.5) such small tumours. We thank the reviewer to actually pointing out this point that was overlooked. In fact, after this correction, the data is even more robust.

Reviewer #3 (Remarks to the Author):

The resubmission from Sanz-Moreno and coauthors considerably strengthens the original submission. The authors have added numerous additional experiments appropriately addressing most of the reviewers' legitimate concerns. They have clarified several key methods that raised questions, addressed questions regarding the influence of proliferation and viability, and made appropriate references to their previous work characterizing the migratory behavior of the amoeboid cells reported here which was a major concern for this reviewer. In short, the rebuttal and associated revised manuscript are thoroughly and thoughtful constructed, and the authors should be commended for their efforts. The finding is novel, well-supported by the experiments, and will be of interest to several audiences. No further revisions are recommended.

We are glad that the Reviewer is happy with the revised manuscript, and again we thank the Reviewer for the comments which have improved our study.

REVIEWERS' COMMENTS:

Reviewer #2 (Remarks to the Author):

The authors have addressed all of my concerns. Congratulations on the excellent manuscript.